

# Machine Learning Model for Inverting Convective Boundary Layer Height with Implicit Physical Constraints and Its Multi-Site Applicability

Yufei Chu[1], Guo Lin[2,3], Min Deng[4], Lulin Xue[5], Weiwei Li[5], Hyeyum Hailey Shin[5], Jun A. Zhang[2,3],
Hanqing Guo[6] , and Zhien Wang[1]

[1]School of Marine and Atmospheric Sciences, Stony Brook University, Stony Brook, 11790, USA
[2]NOAA /AOML/Hurricane Research Division, Miami, 33149, USA.
[3]Cooperative Institute for Marine and Atmospheric Studies, University of Miami, Miami, 33149, USA
[4]Environmental Science and Technologies Department, Brookhaven National Laboratory, Upton,
11793, USA
[5]National Science Foundation National Center for Atmospheric Research, Boulder, 80307, USA
[6]Department of Electrical and Computer Engineering, University of Hawaii at Manoa, Honolulu,96822,
USA

*Correspondence to*: Zhien Wang (Zhien.wang@stonybrook.edu)

**Abstract.**  Accurate estimation of convective boundary layer height (CBLH) is vital for weather, climate, and air quality modeling. Machine learning (ML) shows promise in CBLH prediction, but input parameter selection often lacks physical grounding, limiting generalizability. This study introduces a novel ML framework for CBLH inversion, integrating thermodynamic constraints and the diurnal CBLH cycle as an implicit physical guide. Boundary layer growth is modeled as driven by surface heat fluxes and atmospheric heat absorption, using the diurnal cycle as input and output. TPOT and AutoKeras are employed to select optimal models, validated against Doppler lidar-derived CBLH data, achieving an $R^2$ of 0.84 across untrained years. Comparisons of eddy covariance (ECOR) and energy balance Bowen ratio (EBBR) flux measurements show consistent predictions ($R^2$ difference ~0.011, MAE ~0.002 km). Models trained on ARM SGP C1 site ECOR data and tested at E37 and E39 yield $R^2$ values of 0.787 and 0.806, respectively, demonstrating adaptability. Training with all sites' data enhances C1 ECOR and EBBR performance over C1-only training: ECOR ($R^2$: 0.851 vs. 0.845; MAE: 0.198 km vs. 0.207 km), EBBR ($R^2$: 0.837 vs. 0.834; MAE: 0.203 km vs. 0.205 km). The interquartile range (IQR) error bars for predicted CBLH are consistently narrower than those for DL-derived CBLH, reflecting lower variability in predicted CBLH compared to DL-derived CBLH, which is influenced by additional factors. Transferability across ARM Southern Great Plains sites and seasonal performance during summer (JJA) confirm the model's robustness, offering a scalable approach for improving boundary layer parameterization in atmospheric models.



# 1 Introduction

The convective boundary layer (CBL) is a critical component of the Earth's atmosphere, governing the exchange of heat, moisture, and momentum between the surface and the free troposphere (Stull, 1988; Garratt, 1994). Accurate estimation of the convective boundary layer height (CBLH) is essential for understanding atmospheric processes, including turbulence, pollutant dispersion, and cloud formation (Stull, 1988; Seibert et al., 2000). In numerical weather prediction (NWP) and climate models, CBLH
serves as a key parameter for parameterizing turbulent mixing and convective processes, directly impacting forecast accuracy and climate projections (Grenier, 2001; Holtslag et al., 2013; Baklanov et al., 2014). Errors in CBLH estimation can lead to significant biases in surface temperature, humidity, and air quality predictions (Vogelezang and Holtslag, 1996; Hu et al., 2010). Consequently, improving CBLH predictions has been a priority in atmospheric science, with numerous studies emphasizing its role in
model performance and data assimilation (Helmis, 2012; Cohen et al., 2015; Wulfmeyer et al., 2016; Brown et al., 2008; Barlow et al., 2020; Chu et al., 2022a).

While current observational techniques have greatly contributed to the determination of the CBLH, each method still presents inherent limitations related to resolution, sensitivity, or applicability under different
atmospheric conditions. Radiosondes provide direct measurements of temperature and humidity profiles but suffer from low temporal resolution, typically limited to twice to fourth-daily launches (Seidel et al., 2010; Liu and Liang, 2010; Lin, 2024). Meteorological towers measure near-surface variables but are constrained by their height, rarely capturing the full CBL (Bianco et al., 2011; Emeis et al., 2009). Weather radars offer vertical profiles but lack the resolution to resolve fine-scale CBL structures (Heo et
al., 2003; Compton et al., 2013). Aerosol lidars, while effective for detecting entrainment zones, are often confounded by residual layers, leading to ambiguous CBLH estimates (Hennemuth and Lammert, 2006; Sawyer and Li, 2013; Schween et al., 2014; Luo, 2014). Doppler lidars provide high-resolution velocity and backscatter data, enabling precise CBLH retrievals, but their algorithms vary widely (Tucker et al., 2009; Barlow et al., 2011; Chu et al., 2020). Each method employs different inversion algorithms—such
as gradient-based, variance-based, or wavelet techniques—each with inherent uncertainties depending on atmospheric conditions and data quality (Cohn and Angevine, 2000; Hägeli et al., 2000; Lammert and Bösenberg, 2006; Compton et al., 2013; Chu et al., 2022b).

Recent advances in machine learning (ML) have revolutionized CBLH prediction by leveraging large
datasets to model complex atmospheric relationships. Early ML approaches used simple regression models to estimate CBLH from radiosonde data (Krishnamurthy et al., 2020; Madonna et al., 2021). Subsequent studies adopted random forests and neural networks, incorporating inputs from aerosol lidars, Doppler lidars, and reanalysis datasets (Liu et al., 2022; Krishnamurthy et al., 2021; Peng et al., 2023; Wei et al., 2025; Zhang et al., 2025). For instance, random forest models have been applied to lidar-
derived backscatter profiles (Du et al., 2023; Chu et al., 2025a), while deep neural networks have integrated reanalysis data for regional CBLH predictions (Ayazpour et al., 2023; Su et al., 2024). Despite these advances, most ML models select input parameters empirically, lacking physical constraints, which limits their generalizability across diverse sites (de Arruda Moreira et al., 2022; Su et al., 2024; Chu et al., 2025a; Macatangay et al., 2025; Stapleton et al., 2025). Few studies have explored physically





constrained ML frameworks or evaluated model performance across multiple stations, highlighting a
    critical gap in the literature (Krishnamurthy et al., 2021; Su et al., 2024; Wei et al., 2025; Stapleton et al.,
    2025). The above discussion and existing literature clearly demonstrate that numerous ML algorithms are
    currently available. Comparing each algorithm to identify the optimal one is highly time-consuming.

Building on these insights, this study introduces an Auto-ML framework that automatically selects the
    optimal ML algorithm for CBLH inversion, utilizing Doppler lidar-derived CBLH data and
    thermodynamically constrained input parameters. We incorporate physical principles by assuming CBL
    growth is driven by thermodynamic equilibrium, with surface heat flux and atmospheric heat absorption
    as primary drivers. This approach ensures robust predictions across varying atmospheric conditions and
sites. To assess the model's transferability, we evaluate its performance at four sub-sites (C1, E32, E37,
    and E39) within the Atmospheric Radiation Measurement (ARM) Southern Great Plains (SGP) supersite.
    These locations were selected due to their comprehensive observations of surface heat flux (SHF), latent
    heat flux (LHF), and lower tropospheric stability (LTS). These sites provide a diverse testbed for
    validating the model's generalizability and its potential to enhance CBLH predictions in atmospheric
models.

    This paper is organized as follows: Section 2 describes the data sources and ML methodology, including
    the implicit physical constraints. Section 3 presents the model results, encompassing performance metrics,
    site-to-site comparisons, and contrasts across different seasons and ML approaches. Section 4 discusses
the findings, their implications for atmospheric model, and future research directions.

## 2 Data and methods

### 2.1 Site description

This study utilizes data from the ARM SGP facility, a premier research site established by the U.S.
Department of Energy to investigate land-atmosphere interactions in a continental mid-latitude
environment (Mather et al., 2016). Located in Oklahoma, USA, the SGP spans a diverse agricultural
    landscape, making it ideal for studying CBL dynamics under varying meteorological conditions (Mather
    and Voyles, 2013). We focus on four SGP sites: the central facility (C1) and three extended facilities (E32,
    E37, E39), selected for their comprehensive measurements of surface fluxes and atmospheric profiles.
    The latitude and longitude coordinates of the four sites are shown in Table 1 (Wulfmeyer et al., 2018).
The C1 site, located near Lamont, Oklahoma, serves as the primary hub, hosting a suite of instruments
    including radiosondes, a Doppler lidar, and an Atmospheric Emitted Radiance Interferometer (AERI).
    The extended sites—E37, and E39—are equipped with Eddy Correlation (ECOR) systems for surface
    flux measurements. Additionally, the nearby E14 site (Lamont, Oklahoma; 36.605°N, 97.485°W, 315 m
    elevation), co-located with C1, which we attribute to C1 for consistency. The E32, E39, and E13 (near
C1) sites employ Energy Balance Bowen Ratio (EBBR) technology to measure heat flux.





Table 1. Instruments and datasets used in this study.

| Sites | Latitude (°, N) | Longitude (°, W) | Altitude (m) | AERI data stream | DL data stream | ECOR data stream | EBBR data stream | Other |
|---|---|---|---|---|---|---|---|---|
| C1 | 36.6073 | 97.4876 | 314 | sgpaerioe1turnC1.c1. | sgpdlfptc1.b1 | sgp30qcecorE14.s1. | sgp30baebbrE13.c1. | |
| E32 | 36.8193 | 97.8198 | 328 | sgpaerioe1turnE32.c1. | Sgpdlfpte32.b1 | / | sgp30baebbrE32.c1. | E14's ECOR or E13's EBBR used as C1 |
| E37 | 36.3104 | 97.9274 | 379 | sgpaerioe1turnE37.c1. | Sgpdlfpte37.b1 | sgp30qcecorE37.s1. | / | |
| E39 | 36.3735 | 97.0691 | 279 | sgpaerioe1turnE39.c1. | Sgpdlfpte39.b1 | sgp30qcecorE39.s1. | sgp30baebbrE39.c1. | |


The distances between the ARM SGP sites are as follows: C1 to E32 is approximately 40 km, C1 to E37 is approximately 77km, C1 to E39 is approximately 41 km, E32 to E37 is approximately 57km, E32 to E39 is approximately 67 km, and E37 to E39 is approximately 77 km. These distances ensure a range of spatial variability in surface and atmospheric conditions, enabling robust evaluation of the model's

transferability across sites (Turner et al., 2016). The C1 site's Doppler lidar (DL) provides high-resolution vertical velocity and backscatter data, while radiosondes offer 4-th-daily temperature and humidity profiles. The AERI at four sites measures downwelling infrared radiance to derive atmospheric stability metrics.

## 2.2 Data and preprocessing

The dataset comprises multiple variables critical for CBLH estimation, sourced from the ARM SGP sites over the period 2016–2019. The DL used are Halo Photonics Stream Line models (1.5 μm wavelength), with the C1 site featuring an upgraded Stream Line XR+ model for enhanced signal-to-noise ratio (SNR). These lidars provide a vertical resolution of 30 m and a temporal resolution of 1-3s, ensuring detailed vertical velocity profiles (Manninen et al., 2019). The CBLH is calculated using Chu et al. (2022a)'s

algorithm on ARM DL data, utilizing wavelet analysis to account for turbulence eddy size and gravity wave effects, and applying dynamic thresholds to estimate CBLH from 2-D vertical velocity variance. LTS is derived from AERI observations at C1, calculated as the potential temperature difference between 700 hPa and 1000 hPa (LTS = $\theta_{700} - \theta_{1000}$), validated against radiosonde data (Feltz et al., 2003; Wood et al., 2006). Surface fluxes, including SHF and LHF, are obtained from ECOR systems at C1 (via E14),

E37, and E39, and from the EBBR system at C1 (via E13), E32, and E39. The ECOR systems use eddy covariance techniques to measure turbulent fluxes, while the EBBR system estimates fluxes via the Bowen ratio method, incorporating net radiation, soil heat flux, and temperature-humidity gradients (Cook, 2005; Cook, 2011).

Previous studies have shown significant flux discrepancies between ECOR and EBBR beams obtained through different detection techniques, making them non-interchangeable for direct use (Tang et al., 2019; Chu et al., 2022b). Data preprocessing involves quality control to remove outliers and missing values, following ARM's standard protocols (e.g., flagging data with unrealistic values or low SNR).





### 2.3 Machine learning methods

Machine learning (ML) algorithms have emerged as powerful tools in atmospheric science, enabling the analysis of complex, non-linear relationships within large datasets to improve predictions of phenomena such as CBLH (Schultz et al., 2021). ML methods excel at identifying patterns in atmospheric data, enhancing applications like weather forecasting, air quality modeling, and boundary layer parameterization by integrating diverse data sources, including ground-based observations and reanalysis
products (Reichstein et al., 2019). Two prominent ML approaches for regression tasks are decision tree-based methods and neural networks, each offering distinct advantages for atmospheric applications (Bauer et al., 2015; de Burgh-Day et al., 2023).

Decision tree-based methods partition data into hierarchical decision nodes, creating a flowchart-like
structure to predict outcomes based on input features. Advanced ensemble techniques, such as random forests and gradient boosting, combine multiple trees to improve accuracy and robustness, making them well-suited for tasks like CBLH estimation (Breiman, 2001; Chen and Guestrin, 2016). Neural networks, conversely, consist of interconnected layers of nodes that learn intricate patterns through backpropagation, excelling in capturing non-linear dynamics in atmospheric datasets, such as turbulence or stability
gradients (Goodfellow et al., 2016). Automated ML frameworks streamline model development by optimizing architectures and hyperparameters. Prior studies have compared frameworks like AutoKeras (Zhong et al., 2024), which automates neural network design, and the Tree-based Pipeline Optimization Tool (TPOT), which focuses on tree-based models, finding comparable performance in atmospheric applications (Jin et al., 2019; Olson et al., 2016). Considering computational efficiency and the
adaptability of algorithms to diverse datasets, this study does not simultaneously compare the results of various machine learning methods. Instead, it focuses on comparing the outcomes of TPOT and AutoKeras after their automated selection of optimal models. Specifically, we adopt the TPOT Regressor from the TPOT library, which automates the construction and optimization of tree-based pipelines, minimizing manual tuning efforts while preserving high predictive accuracy (Olson et al., 2016).
Similarly, we employ AutoKeras, a neural architecture search framework that leverages Bayesian optimization and network morphism to automatically design and fine-tune deep learning models, enabling efficient and adaptive model selection for complex datasets (Jin et al., 2019; Liang et al., 2024). This research employs the automated machine learning frameworks TPOT (version 0.12.2) and AutoKeras (version 1.0.20), integrated within a Python 3.9 environment. Development was executed using the
spyder-kernels package (version 2.4.4), ensuring robust and reproducible computational workflows. To facilitate reader understanding, the table 2 below provides an overview of commonly supported algorithms in TPOT and AutoKeras, along with their descriptions, advantages, applicable scenarios, disadvantages, and limitations, supported by relevant references.






Table 2. Commonly supported algorithms in TPOT and AutoKeras.

| Framework | Algorithm | Description | Advantages | Limitations | Reference |
|---|---|---|---|---|---|
| **TPOT** | Decision Tree Regressor | A tree-based model that splits data into branches based on feature thresholds to predict outcomes. | Simple, interpretable; suitable for small-to-medium datasets with clear feature relationships. | Prone to overfitting; struggles with high-dimensional or noisy data. | Breiman et al. (1984) |
| | Random Forest Regressor | An ensemble of decision trees that aggregates predictions to reduce variance and improve accuracy. | Robust to overfitting, handles non-linear relationships well; ideal for medium-sized datasets. | Computationally intensive; less interpretable due to ensemble nature. | Breiman (2001) |
| | Gradient Boosting Regressor (XGBoost) | Boosts weak decision trees sequentially, optimizing a loss function using gradient descent. | High predictive accuracy, handles missing data well; effective for structured/tabular data. | Requires careful hyperparameter tuning; can be slow to train on large datasets. | Chen and Guestrin (2016) |
| | Support Vector Regressor (SVR) | Finds a hyperplane that best fits the data within a margin, using kernel tricks for non-linearity. | Effective for small datasets with non-linear patterns; robust to outliers. | Scales poorly with large datasets; sensitive to kernel choice and parameter tuning. | Drucker et al. (1997) |
| | Linear Regression | Models the relationship between features and target using a linear equation. | Simple, interpretable, fast to train; suitable for datasets with linear relationships. | Assumes linearity and independence of features; performs poorly with non-linear or noisy data. | Seber and Lee (2003) |
| **AutoKeras** | Convolutional Neural Network (CNN) | A deep learning model with convolutional layers to extract spatial features from data. | Excels in image or spatial data processing; automatically learns hierarchical features. | Requires large datasets and computational resources; less interpretable. | LeCun et al. (1998) |
| | Recurrent Neural Network (RNN) | A deep learning model designed for sequential data, with loops to retain memory of past inputs. | Suitable for time-series or sequential data; captures temporal dependencies. | Prone to vanishing gradients; computationally expensive for long sequences. | Hochreiter and Schmidhuber (1997) |
| | Transformer | A model using self-attention mechanisms to process sequential data, often for time-series tasks. | Efficient for long sequences, captures global dependencies; ideal for complex time-series data. | High computational cost; requires large datasets for effective training. | Vaswani et al. (2017) |
| | Multilayer Perceptron (MLP) | A fully connected neural network with multiple layers to model non-linear relationships. | Versatile for tabular data; can approximate complex functions with sufficient depth. | Prone to overfitting on small datasets; requires careful tuning of architecture and parameters. | Hornik et al. (1989) |
| | Residual Network (ResNet) | A deep CNN with residual connections to mitigate vanishing gradients in deep architectures. | Enables training of very deep networks; effective for complex pattern recognition tasks. | High computational cost; may overfit on small datasets without proper regularization. | He et al. (2016) |

## 2.4 Evaluation metrics

To assess the performance of the ML model, we use a suite of standard regression metrics, providing a comprehensive evaluation of predictive accuracy and error characteristics. Let $y_i$ represent the observed
CBLH, $\widehat{y_i}$ the predicted CBLH, and $n$ the number of observations. The following metrics are used:

**Coefficient of Determination ($R^2$):** Measures the proportion of variance in the observed CBLH explained by the model, calculated as:

$$R^2 = 1 - \frac{\sum_{i=1}^{n}(y_i - \widehat{y_i})^2}{\sum_{i=1}^{n}(y_i - \bar{y})^2}, \qquad\qquad (1)$$





where $\bar{y} = \frac{1}{n}\sum_{i=1}^{n} y_i$ is the mean of the observed values (Krause et al., 2005; Legates and McCabe, 1999). The coefficient of determination ($R^2$) quantifies the model's explanatory power by measuring the proportion of variance in the observed CBLH that is accounted for by the model predictions. As shown in Equation (1), $R^2$ compares the sum of squared residuals (SSR) against the total variance in the observations (SST). When $R^2$ approaches 1, it indicates that the model explains nearly all variability in

the observed CBLH (SSR ≈ 0), representing a near-perfect fit where predicted values ($\hat{y}_i$) closely match observations ($y_i$). Conversely, an $R^2$ near 0 suggests the model performs no better than simply using the mean ($\bar{y}$) as a predictor. While useful for goodness-of-fit assessment, we need to notice that high $R^2$ values alone do not guarantee model validity, as they may be artificially inflated by overfitting or insensitive to systematic biases in the predictions.


**Mean Absolute Error (MAE):** Quantifies the average absolute difference between predicted and observed CBLH, given by:

$$MAE = \frac{1}{n}\sum_{i=1}^{n} |y_i - \hat{y}_i| \ , \tag{2}$$

MAE provides a straightforward measure of prediction error in the same units as CBLH (Willmott and
Matsuura, 2005). MAE measures the average absolute difference between predicted and true values. It is used to monitor the model's performance during training and validation but does not directly influence the optimization of the model parameters (i.e., it does not affect the minimization of the loss function). MAE provides an intuitive metric to reflect the average magnitude of prediction errors.

**Mean Squared Error (MSE):** Computes the average squared difference between predictions and observations, defined as:

$$MSE = \frac{1}{n}\sum_{i=1}^{n} (y_i - \hat{y}_i)^2 \ , \tag{3}$$

The quadratic nature of Mean Squared Error (MSE) inherently penalizes larger prediction errors more severely (Hyndman & Koehler, 2006). AutoKeras optimizes model parameters by directly minimizing
this loss function to enhance predictive accuracy. In contrast, TPOT utilizes Negative MSE (−MSE), a transformed version where the sign inversion converts the minimization problem into a maximization framework. This approach maintains consistency with scikit-learn's convention where higher scores denote superior models, allowing TPOT's automated search to effectively identify configurations that maximize -MSE (thereby minimizing the actual MSE).


This study primarily focuses on comparing the CBLH across different sites, utilizing the $R^2$ and MAE as robust metrics to evaluate the performance of machine learning models. These metrics provide a comprehensive assessment of model accuracy and predictive reliability, with $R^2$ quantifying the proportion of variance in CBLH that is explained by the model, and MAE offering a direct measure of
the average prediction error in physical units. These metrics collectively ensure a thorough evaluation of the model's accuracy, bias, and robustness across the ARM SGP sites.





## 2.5 Implicit physical constraints

### 2.5.1 Parameter selection based on thermodynamic equilibrium constraints

Traditional machine learning methods for predicting CBLH typically employ Principal Component Analysis (PCA) or random combinatorial approaches to select input parameters (Liu et al., 2022). While these methods can achieve good predictive performance at specific sites, they lack a physical basis, resulting in poor transferability across different sites and limiting their applicability in physical space. To address this issue, this study proposes an innovative approach by incorporating the physical foundation

of thermodynamic equilibrium to optimize parameter selection, thereby developing a CBLH prediction model that is transferable across sites. As noted by Stull (1988) in his classic textbook, although the development of the convective boundary layer is influenced by multiple factors, thermodynamic equilibrium is the primary driver of its evolution. Figure 1a illustrates the heat required for CBL growth, while Fig.1b depicts the surface heat flux emitted following solar radiation absorption,   which together

determine the CBLH. Additionally, this study builds on the theoretical framework of thermodynamic equilibrium to quantify the dynamic evolution of the CBLH. Specifically, the relationship is described by the following integral expression:

$$\int_{t=0}^{t_1} \overline{w'\theta_s'(t)}dt = \int_{\theta=\theta_0}^{\theta_1} Z(\theta)d\theta \ , \tag{4}$$

where the left-hand side represents the cumulative contribution of surface heat flux over time, and the

right-hand side describes the amount of heat required to produce the observed temperature profile variation with height within the boundary layer. This heat conservation relationship jointly determines the dynamic evolution of CBLH. This formula provides a solid physical foundation for the model, ensuring that parameter selection not only enhances predictive capability but also maintains physical consistency across different sites (e.g., C1, E32, E37, E39), thereby overcoming the transferability

limitations of traditional machine learning approaches. By integrating the thermodynamics information of the sites, the model can effectively capture the variations in thermodynamic processes across different sites, thus improving both prediction accuracy and generalizability.

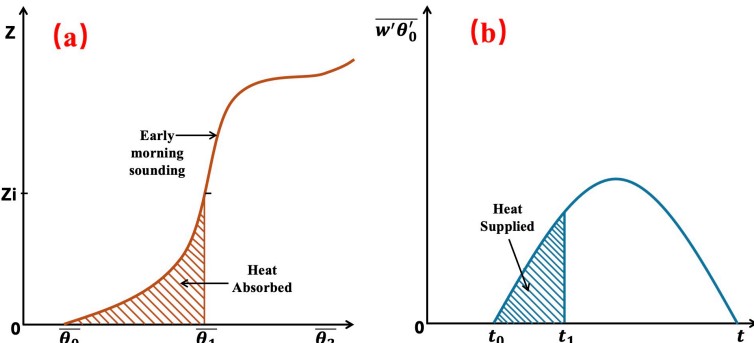

**Figure 1. Graphical approach to estimate Convective Boundary Layer Height (CBLH) thermodynamically by equating heat supplied with heat absorbed; Zi is CBLH.  (Stull, 1988)**



Based on the thermodynamic constraints outlined, traditional physical constraint methods can incorporate the thermodynamic constraints from Equation 4 into the loss function of machine learning models. This

approach adds to the MSE the difference between the cumulative energy provided by the heat flux and the energy required to account for the atmospheric temperature difference, multiplied by an adjustment coefficient to optimize the model. However, the selection of this coefficient significantly impacts the results, posing challenges to the method's applicability. From the perspective of boundary layer physics, Equation 4 provides a robust physical constraint during the development from sunrise to the top of the

CBL. Nevertheless, after reaching the boundary layer top, the entrainment process in the entrainment zone must also be considered, where the application of Equation 4 may impose inappropriate constraints. The dissipation process, on the other hand, involves distinct physical mechanisms. Additionally, factors such as moisture, wind speed, wind direction, and cloud cover introduce complex, nonlinear effects, making the direct application of physical constraints highly challenging.


This study proposes an innovative approach that combines heat flux with LTS to achieve implicit physical constraints. Initially, the cumulative heat flux is decomposed into SHF and LHF, which reflect moisture characteristics. Furthermore, instantaneous SHF and LHF can account for the influences of clouds and wind. Simultaneously, LTS is calculated as an instantaneous value for each hour, dynamically capturing

temperature difference variations. To address diurnal and seasonal variations in CBLH, the model incorporates sunrise and sunset times along with their corresponding timestamps, defining a normalized temporal parameter,

$$\mathrm{SUNPERCENT} = (\mathrm{TIME} - \mathrm{SUNRISE}) / (\mathrm{SUNSET} - \mathrm{SUNRISE}) \ , \tag{5}$$


which represents the proportion of the current time relative to the daylight duration. In summary, this study employs physically driven variables for parameter selection—specifically surface heat flux and LTS as core inputs—to ensure the model's physical consistency and transferability. The heat flux is further broken down into physical components, including the cumulative sensible heat flux (C_SHF) and

latent heat flux (C_LHF) since sunrise, as well as the instantaneous sensible heat flux (I_SHF) and latent heat flux (I_LHF) within a one-hour window, while LTS is taken as an hourly instantaneous value. This parameterization effectively captures diurnal variations in solar radiation, enriching the model with more comprehensive physical information.

### 2.5.2 Integrated Diurnal Evolution of the CBL


The current literature on predicting CBLH using machine learning predominantly focuses on discrete, moment-to-moment predictions, often overlooking the integrated diurnal evolution of the CBL as a unified process. For instance, Chu et al. (2025a) employed ML to estimate CBLH over the Southern Great Plains, but this approach centered on individual time steps, neglecting the full diurnal cycle. However, as

shown in Fig. 2, the diurnal variation of CBLH across five ARM sites reveals distinct site-specific patterns. The CBLH at each moment evolves continuously from the preceding moment, establishing a dynamic and interconnected developmental trajectory. Treating these moments in isolation disrupts this continuity,



failing to capture the underlying evolutionary dynamics. Specifically, the peak CBLH values vary across
the sites, and the morning growth and evening decay phases exhibit notable differences, highlighting the
critical role of temporal dependencies in boundary layer evolution. While some studies incorporate the
CBLH of the previous moment—or CBLH derived from alternative methods, such as sensible heat flux
or parcel methods—as an input variable, this approach often overemphasizes the influence of prior CBLH
values, thereby overshadowing the contributions of other meteorological drivers. For example, Su et al.
(2020) demonstrated that machine learning models relying heavily on CBLH derived from sensible heat
flux and parcel methods tend to exhibit excessive dependence on temporal autocorrelation, which
diminishes the model's sensitivity to key meteorological factors such as heat flux and atmospheric
stability. Consequently, these methods are limited in their ability to comprehensively predict the diurnal
variation of CBLH, constraining the scope of their investigations.

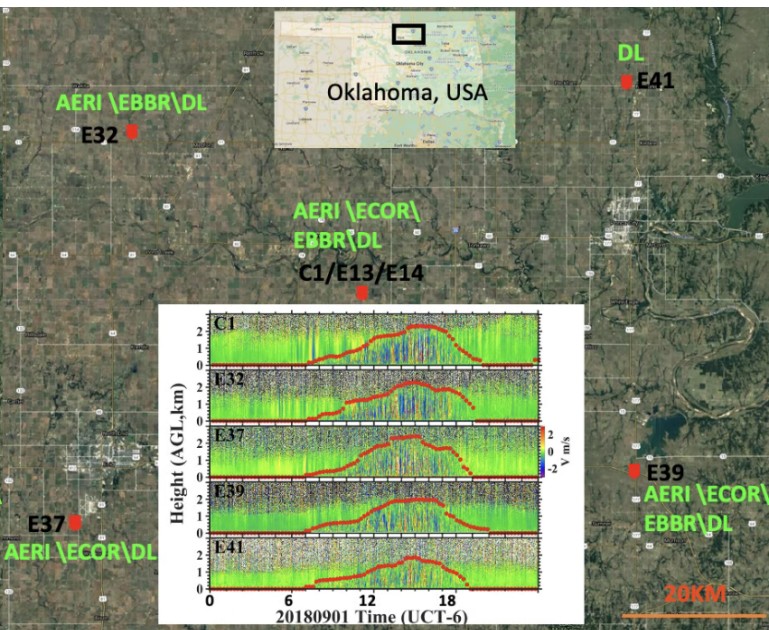


**Figure 2. Relative Locations of ARM SGP Sites C1, E32, E37, E39, and E41. Top inset: Geographic position of SGP sites in
Oklahoma; Bottom inset: Diurnal variation of CBLH observed by Doppler lidar across the five sites on 01 September 2018.**

To address these shortcomings, this study adopts the CBLH across the entire diurnal cycle as the training
target for the machine learning model, treating the boundary layer evolution as a continuous and
interconnected process. This holistic approach enables the model to comprehensively capture the dynamic
evolution of the boundary layer, from the gradual rise of the CBLH after sunrise, through its midday peak
accompanied by oscillations, to the rapid decay observed after sunset. By integrating the complete
developmental trajectory, the model not only better represents the interconnected dynamics of the CBL
but also accounts for the complex interplay of meteorological drivers that govern its evolution. For
instance, the morning growth phase is heavily influenced by surface heating and turbulent mixing, while
the midday peak often reflects a balance between entrainment processes at the boundary layer top and
surface-driven convection. The evening decay phase, on the other hand, is modulated by radiative cooling





and the cessation of surface heat fluxes, which vary significantly across different sites due to local land
surface characteristics and atmospheric conditions. To enhance the model's predictive capability, we
incorporate time-dependent variables that reflect the diurnal cycle's progression. This approach mitigates
the overreliance on prior CBLH values by ensuring that the model learns the underlying physical
relationships between CBLH and its meteorological drivers, rather than simply exploiting temporal
autocorrelation. As a result, the model is expected to improve the accuracy of CBLH predictions across
the diurnal cycle, offering a more comprehensive understanding of boundary layer dynamics.

## 2.6 Auto-ML model for convective boundary layer height

We prepared the relevant input parameters and employed the following methodology to enable the
machine learning approach to uncover the complex physical mechanisms underlying the physical
parameters. After the compilation environment was set up and the data was prepared, the specific model
application process (Fig. 3) is as follows:

1)**Data Collection and Pre-processing**. Collect and pre-process the necessary input data for CBLH
prediction. Based on the content of Section 2.1 and 2.2, we prepare the data for each timestamp of the
day, including C_SHF, C_LHF, I_SHF, I_LHF, LTS, TIME, SUNRISE, SUNSET, SUNPERCENT, and
CBLH. The SUNRISE and SUNSET represent the sunrise and sunset times calculated based on the
latitude and longitude coordinates of the site. To simplify the dataset, we aggregate the data from 06:00
to 21:00 (UTC-6), covering a 15-hour period, as a single daily dataset. The CBLH for the entire day is
designated as the target variable for output, while the other parameters serve as input variables. We
randomly split all the data into 70% for training and 30% for testing by date. The subfigure in Fig. 3 that
depicts the ARM site, included in the Data Collection section, is adapted from Wulfmeyer et al. (2018).

2)**Use AutoML to find the Best Train Model**. Using the training dataset, we employ TPOT and
AutoKeras to derive their respective optimal algorithms or hyperparameters. By comparing the R² and
Mean MAE metrics, we select the model that performs best in both MAE and R² as the optimal model,
which is then designated as the candidate best model for further evaluation and application.

3)**Use the Best Model with training Data for Training.** The best model is trained on the training
dataset and saved for later use, ensuring optimal performance for future applications.
4)**One-Day CBLH Prediction.** Use the trained model to predict CBLH for a single day.

5)**Check if All Days are Processed.** If not all days in the test dataset have been processed, repeat step
4 for the next day.
6)**Save Data.** When all days' data have been processed, save the predicted CBLH data for further
analysis.





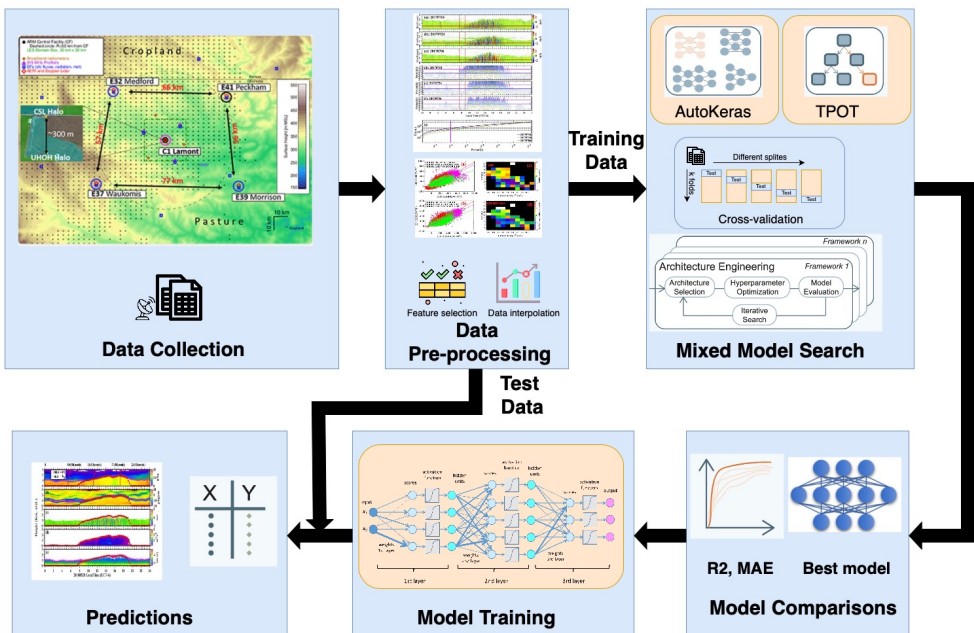

**Figure 3. The Auto-ML workflow of Convective Boundary Layer Height (CBLH)**

Figure 3's flowchart outlines the algorithm proposed in this study (termed the Auto-ML algorithm). Its core principles are: (1) utilizing thermodynamical variables as input parameters with implicit physical constraints; (2) incorporating the complete CBL development cycle as unified input, with corresponding CBLH as output; and (3) employing TPOT and AutoKeras models to automatically select the optimal machine learning algorithm. This approach enables the model to capture the entire CBL development 375 process, enhancing prediction accuracy and representation of CBL dynamics.

## 3 Results

To validate the effectiveness of the Auto-ML framework, we first conducted tests using data from the C1 site spanning 2016 to 2019, presenting the results for ECOR and EBBR heat flux, respectively. Subsequently, the algorithm was applied to model data transfer from one site to other sites. Next, we 380 compared the performance of the optimal TPOT and AutoKeras algorithms for summer (JJA) and further evaluated the advantages and limitations of different methods for computing SHAP (Shapley Additive exPlanations) values. Furthermore, we analyzed the variations in Auto-ML's relative importance across seasons. Finally, we compared the performance of models trained on multi-site data and tested on site-specific data.





## 3.1 Application of the Auto-ML to the ARM SGP C1 Site

### 3.1.1 Results from ECOR flux

The Auto-ML algorithm demonstrates robust performance in predicting CBLH when using ECOR flux dataset. The scatter predicted CBLH in Fig.4a demonstrates a strong linear correlation ($R^2$ = 0.845) between predicted and observed CBLH across the annual dataset, suggesting that the Auto ML model effectively captures the general trends of CBL evolution (Fig.4a). However, the MAE of 0.207 km highlights a non-negligible systematic bias, potentially linked to specific meteorological conditions or seasonal variations not fully resolved by the model. Notably, the density of points clusters around the 1:1 line in the lower CBLH range (0–1.5 km), while deviations increase slightly at higher CBLH values (>2 km), possibly indicating reduced model sensitivity to extreme events (e.g., intense convective days).

The diurnal variability between predicted and observed CBLH also shows good agreement (Fig.4b). The predicted CBLH closely tracks observed values during the morning development phase (07:30–13:30 UTC-6), with overlapping interquartile ranges (IQR error bars), reflecting reliable performance during periods of rapid boundary layer growth driven by surface heating and turbulent mixing. However, a significant divergence emerges in the afternoon (15:30–17:30 UTC-6), where the predicted mean CBLH underestimates observations by ~100 m. This discrepancy coincides with the typical peak phase of the CBL, characterized by weakening turbulence, entrainment processes at the CBL top, and increasing influence of subsidence or advection.

The afternoon underestimation may stem from the algorithm's limited ability to resolve complex interactions during the CBL peak phase. During midday, solar radiation maximizes surface heat flux, driving vigorous turbulent eddies that homogenize the CBL, making CBLH prediction relatively straightforward. By late afternoon, surface heating diminishes, turbulence decays, and the entrainment zone at the CBL top becomes dynamically significant. Entrainment of free-tropospheric air into the CBL can temporarily elevate the observed CBLH, a process that may not be easily captured in the Auto ML model. Additionally, the advection of air masses with different thermodynamic properties (e.g., moisture or temperature gradients) could introduce spatial heterogeneity, further challenging the algorithm's generalizability during transitional periods.

Moreover, the model's training data might underrepresent late-afternoon scenarios, where PBL dynamics are influenced by mesoscale phenomena (e.g., cloud cover, or topographic effects). For instance, enhanced subsidence or cloud shading at 15:30–17:30 (UTC-6) could suppress turbulent mixing, leading to a shallower predicted CBLH compared to observations.





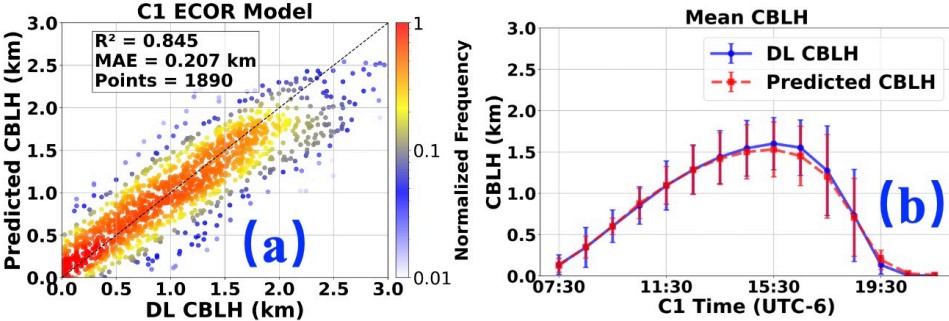

**Figure 4. Results of Auto ML model with ECOR dataset for predicting CBLH: (a) Comparison of all test data, (b) Diurnal variation average with IQR (interquartile range) error bars.**

### 3.1.2 Results from EBBR flux

The Auto-ML algorithm exhibits robust performance in predicting CBLH using the EBBR flux dataset. As shown in Fig. 5a, the scatter plot of predicted CBLH demonstrates a strong linear correlation ($R^2$ = 0.834) between predicted and observed CBLH across the annual dataset, indicating that the Auto-ML model effectively captures the overall trends of CBL evolution. However, the MAE of 0.205 km, comparable to that obtained with the ECOR dataset, suggests that both flux datasets yield similar predictive accuracy. The diurnal variability between predicted and observed CBLH also shows good agreement (Fig.5b). During the morning development phase (07:30–11:30 UTC-6), the predicted CBLH closely aligns with observed values. However, a non-notable divergence emerges in the afternoon (around 15:30 UTC-6), where the predicted CBLH is lower than the observed values, with the deviation reducing from 100 m (as seen with ECOR) to approximately 50 m.

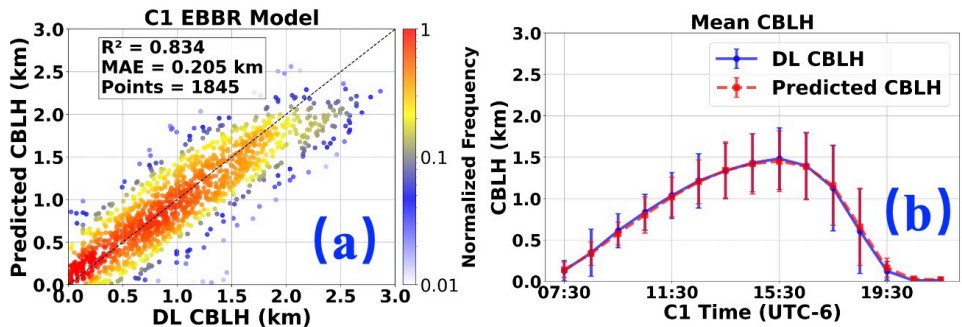

**Figure 5. Results of Auto ML model with EBBR dataset for predicting CBLH: (a) Comparison of all test data, (b) Diurnal variation average with IQR (interquartile range) error bars.**

By comparing Fig.4 and Fig.5, it is evident that, with the same Auto-ML model, the CBLH predictions using ECOR flux data slightly outperform those using EBBR flux data, though the difference is minor ($R^2$: 0.845 vs 0.834; MAE: 0.207 km vs 0.205 km). This suggests that heat flux measurements from different instruments exhibit discrepancies, but these discrepancies follow a systematic, inherent pattern.




The Auto-ML model effectively mitigates these inherent differences, achieving comparable CBLH
predictions, thereby demonstrating its robustness and adaptability.

A comparison of Fig.4 and Fig.5 reveals that the discrepancies between the predicted values and the DL
observations are primarily evident at the PBL top (~15:30 UTC-6) and during the dissipation phase
(~19:30 UTC-6). Specifically, at the CBL top (~15:30 UTC-6), the predicted values are generally lower
than the DL observations, whereas during the dissipation phase (~19:30 UTC-6), the predicted values
tend to exceed the observed values.

### 3.2 Effectiveness of the Auto-ML Across Multiple Sites

The Auto-ML algorithm demonstrates notable adaptability beyond the C1 site, highlighting its potential
for broader application across multiple observation sites. To evaluate this, we tested an Auto-ML model
trained on ECOR heat flux data at the C1 site (E14 site) for its performance at the E37 and E39 sites.
Similarly, an Auto-ML model trained on EBBR heat flux data at the C1 site (E13 site) was assessed for
its performance at the E32 and E39 sites.

### 3.2.1 Apply C1 site ECOR model to E37 and E39 Sites

The Auto-ML model trained at the C1 site with ECOR data exhibits strong performance at the E37 and
E39 sites, achieving $R^2$ values of 0.787 and 0.806, and MAE values of 0.219 km and 0.208 km,
respectively (Fig. 6a and 6c). However, as observed in Fig. 6b and 6d, the model's performance varies
across different time periods at these sites. Specifically, Fig.6b shows that at the E37 site, the model
predictions align well with observations during the CBLH dissipation phase (15:30–21:30 UTC-6).
However, during the initial development phase (07:30–14:30 UTC-6), a significant discrepancy is
observed, with predicted values consistently higher than the DL observations. Notably, no similar
discrepancy is evident in Figure 4b, suggesting that the heat flux differences between the C1 and E37
sites are more pronounced during the initial phase, while the differences diminish after reaching the
boundary layer top.
Similarly, Figure 6d indicates that at the E39 site, the model predictions closely match observations during
the CBLH dissipation phase (15:30–21:30 UTC-6). However, discrepancies are observed during the
initial phase (09:30–11:30 UTC-6) and the top phase (14:30–16:30 UTC-6). During the initial phase, the
predicted values are higher than the DL observations, while at the top phase, the predicted values are
475 lower. In comparison, Figure 4b shows a similar discrepancy at the top phase but not during the initial
phase. This suggests that the heat flux differences between the C1 and E39 sites are more significant
during the initial phase, with the differences decreasing after reaching the boundary layer top.




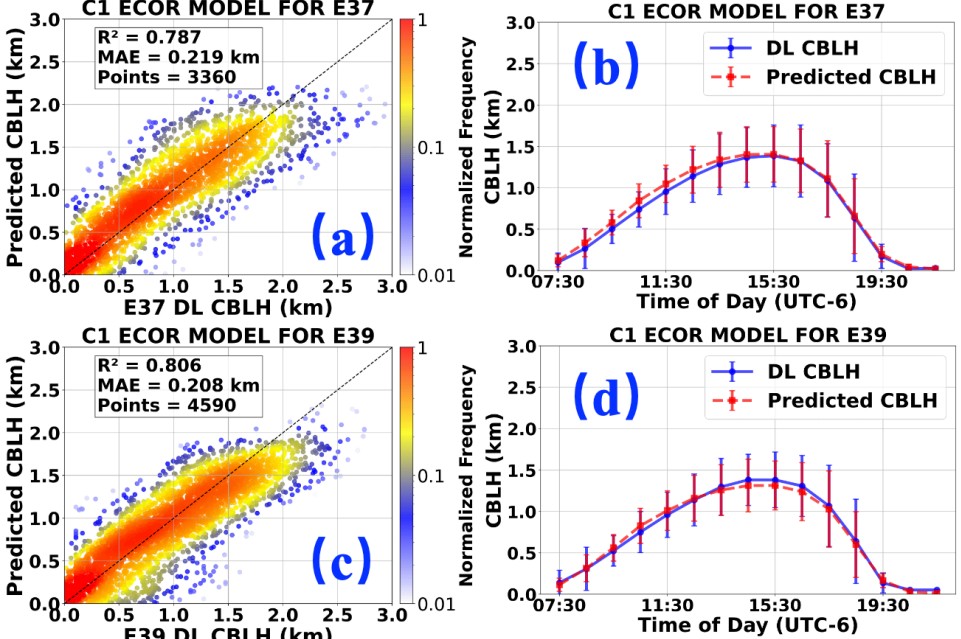

**Figure 6. C1 ECOR model for E37 and E39 sites; (a) and (c) represent the R² and MAE for E37 and E39 respectively; (b) and (d) show the MEAN CBLH with IQR error bars for E37 and E39 respectively.**

The above analysis indicates that while the Auto-ML model trained at the C1 site performs well at the E37 and E39 sites, its performance varies across different time periods, reflecting both similarities and differences in behavior at these sites. This highlights the spatial variability between sites. Furthermore, the differences between the C1 and E39 sites are smaller than those between the C1 and E37 sites, which aligns with their relative distances (41 km vs. 77 km).

### 3.2.2 Apply C1 site EBBR model to E32 and E39 Sites

The Auto-ML model trained at the C1 site using EBBR data performs well at the E39 site, achieving an R² of 0.795 and an MAE of 0.211 km (Fig. 7a), but shows a slight decline in performance at the E32 site, with an R² of 0.719 and an MAE of 0.269 km (Fig. 7c). However, Figure 7d reveals discrepancies in the predicted values near the CBL top, which closely resemble the performance of the C1 site with EBBR data (Fig. 5b). In contrast, Figure 6b indicates significant deviations at the E32 site during both the initial and top phases of CBLH (07:30–17:30 UTC-6), where the predicted values exceed the DL observations by up to ~200 m. Notably, no such pronounced discrepancies are observed in Fig. 5b, suggesting that the heat flux differences between the C1 and E32 sites are substantial during the initial phase and persist even after reaching the boundary layer top.

The above discussion highlights that while the Auto-ML model trained at the C1 site performs effectively at the E37 and E39 sites, it exhibits more pronounced differences at the E32 site. The variability between the C1 and E39 sites is smaller than that between the C1 and E32 sites, despite their comparable distances (41 km vs. 40 km).



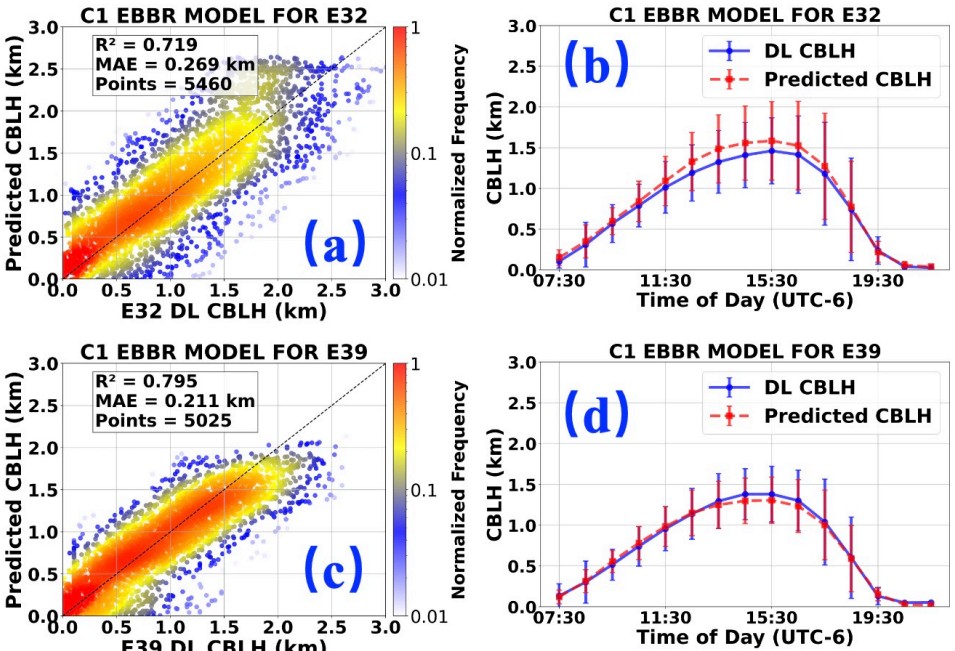

**Figure 7. C1 EBBR model for E32 and E39 sites; (a) and (c) represent the R² and MAE for E32 and E39 respectively; (b) and (d) show the MEAN CBLH with IQR error bars for E32 and E39 respectively.**

The Auto-ML algorithm demonstrates significant adaptability beyond the C1 site, underscoring its
potential for wider application across multiple observational stations (Figs. 6–7). The Auto-ML model
trained at the C1 site performs effectively at the E37 and E39 sites, while models trained at the E37 and
E39 sites also exhibit robust performance at the C1 site, achieving R² values of approximately 0.80–0.85
and MAE values ranging from 0.19 to 0.23 km (figures not shown in this study).

### 3.3 The relationship between Auto-ML performance and the spatial separation between sites

Section 3.2 demonstrates the cross-site applicability of the Auto-ML algorithm. To further investigate the
relationship between Auto-ML performance and the spatial separation between sites, we selected data
from the C1 site (E14 and E13 sites with ECOR and EBBR data, respectively) and the E39 site, training
separate models using both datasets. These models were then applied to other sites with similar heat flux
characteristics to assess the correlation between performance and inter-site distance.
The predictive performance of the Auto-ML algorithm exhibits a clear negative correlation with the
spatial separation between observational sites (Fig. 8). For instance, as shown by the solid red and dashed
lines in Fig.8, the model trained on ECOR data from the C1 site performs robustly at its primary site (R²
= 0.847, MAE = 0.203 km), but its accuracy decreases at the E39 site 41 km away (R² = 0.809, MAE =
0.206 km) and further declines at the E37 site 77 km away (R² = 0.801, MAE = 0.213 km). Similarly, as
depicted by the solid blue and dashed lines in Fig. 8, the model trained on ECOR data from the E39 site
achieves strong performance at its primary site (R² = 0.810, MAE = 0.216 km), but its accuracy higher at
the C1 site 41 km away (R² = 0.836, MAE = 0.208 km) and decreases further at the E37 site 77 km away
(R² = 0.747, MAE = 0.238 km).





Despite this overall trend of performance decline with distance, notable irregularities are observed. For example, the model trained on EBBR data from the C1 site performs best at its primary site ($R^2 = 0.834$, MAE = 0.205 km), with a slight decrease in accuracy at the E39 site 41 km to the southeast ($R^2 = 0.795$, MAE = 0.2011 km), but a more significant decline at the E32 site 40 km to the northwest ($R^2 = 0.719$, MAE = 0.269 km). Likewise, the model trained on EBBR data from the E39 site excels at its primary site

($R^2 = 0.800$, MAE = 0.221 km), maintains comparable performance at the C1 site 41 km to the southeast ($R^2 = 0.805$, MAE = 0.229 km), but shows a substantial drop at the E32 site 67 km to the northwest ($R^2 = 0.758$, MAE = 0.257 km).

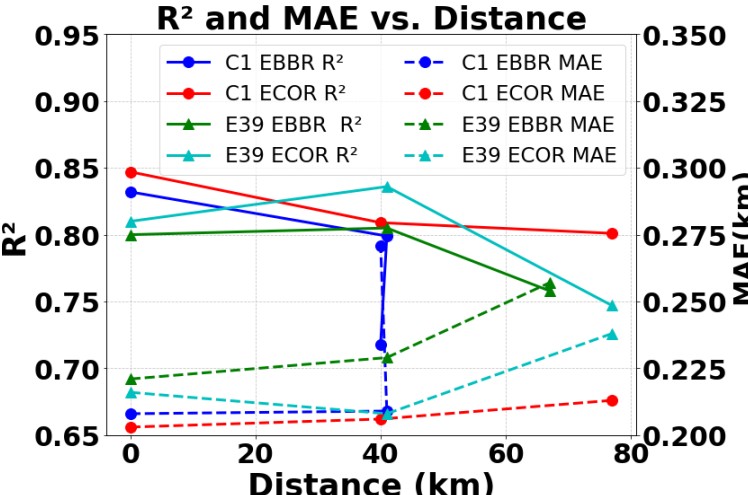

**Figure 8: Relationship between Auto-ML effectiveness and distance.**

It is noteworthy that the model trained on C1 EBBR data exhibits a sharp decline in $R^2$ to 0.71 at the E32 site, located 40 km away, while maintaining a robust $R^2$ of approximately 0.8 at the E39 site, 41 km from C1. One potential reason is that ARM's ECOR systems are typically surrounded by winter wheat fields or farmland, whereas EBBR systems are primarily deployed in pastures. The performance drop at E32

may stem from vegetation differences and measurement principles. E32's pasture dominated EBBR data, prone to overestimating latent heat flux, contrasts with the winter wheat fields around C1 and E39, likely measured by ECOR, which directly captures turbulent fluxes. These discrepancies in surface heat flux inputs challenge the model's generalization, particularly at E32, where site-specific factors like soil moisture or EBBR measurement errors near sunrise/sunset may further degrade performance.


This analysis reveals that the predictive performance of the Auto-ML algorithm exhibits a clear negative correlation with spatial separation between sites, accompanied by spatial heterogeneity. These irregularities align with theoretical expectations: local factors such as terrain variations (e.g., changes in elevation or surface roughness), land use differences (e.g., urban vs. rural settings), and microclimate

effects (e.g., humidity or temperature gradients) disrupt the coherence of CBL dynamics with increasing distance. These site-specific perturbations limit the algorithm's generalizability across diverse regions.



### 3.4 Comparison of Performances of two ML methods for summer at the C1 site.

To compare algorithmic performance, this study evaluates two high-performing machine learning models:
an ExtraTreesRegressor from TPOT and a neural network from AutoKeras. Both models are trained on
the same dataset. The ExtraTreesRegressor is selected in this section because of its inherent resistance to
overfitting. This robustness is achieved through randomized feature selection and split point selection. As
a result, the model performs well on high-dimensional data and noisy datasets and shows strong resilience
to outliers. However, it is not suitable for small-sample datasets. The best-performing model selected by
AutoKeras is a neural network with 10,836 parameters, implemented using the Functional API. It
comprises an input layer for 9-dimensional features, preprocessing layers for multi-category encoding
and normalization (19 non-trainable parameters), two dense hidden layers with 256 and 32 units
respectively (ReLU activation, 10,817 trainable parameters), and a regression output layer. The
architecture leverages AutoKeras's automated feature engineering through integrated preprocessing,
while its two-layer structure maintains moderate complexity. The parameter distribution (256 to 32 units)
indicates a progressive reduction in feature dimensionality, supporting effective feature extraction for the
regression task.

### 3.4.1 SHAP Computation Methods: Tree-Based vs. Gradient Approaches.

To compare the relative importance of features between the two methods, SHAP values for the
ExtraTreesRegressor are computed directly using the TreeExplainer, which leverages the tree structure
(split paths and leaf node values). In contrast, the AutoKeras neural network employs the
GradientExplainer, a gradient-based method, to estimate SHAP values. Their summer (JJA) performance,
shown in Fig. 9, reveals similar $R^2$ and MAE values: ExtraTreesRegressor (0.855, 0.221 km) versus neural
networks (0.840, 0.245 km), as depicted in Fig. 9a (ExtraTreesRegressor) and Fig.9e (neural networks).
These consistent metrics highlight the robustness of both approaches in capturing CBLH. However,
despite their similarity in overall performance, the two models diverge significantly (SHAP method) in
their assessment of feature importance. In Fig.9c, ExtraTreesRegressor assigns a notably higher
importance to LTS (~0.23); attributes nearly equal importance to I_SHF, I_LHF, TIME, and
SUNPERCENT each hovering around 0.15, indicating a clear prioritization of LTS in its decision-making
process. In contrast, the neural network, as shown in Fig.9g, assigns a notably higher importance to I-
LHF (~0.28); attributes nearly equal importance to C_SHF, and I_SHF, each hovering around 0.2. The
different machine learning models can achieve comparable accuracy by using varied nonlinear
combinations of predictors. In such scenarios, the physical interpretation of these models becomes
challenging or may lack sufficient reliability.
Figures 9b and 9d show that the diurnal variations predicted by the AutoKeras neural network and the
TPOT ExtraTreesRegressor are generally comparable. However, Figure 9f reveals that the neural network
predicts lower CBLH values than the ExtraTreesRegressor (Figure 9b) between 07:30–11:30 and around
19:30. Despite this, the neural network exhibits inferior performance, with lower $R^2$ and higher MAE
compared to the ExtraTreesRegressor.





A comparison of Figures 9g–h (neural network) with Figures 9c–d (ExtraTreesRegressor) highlights distinct differences in feature contributions. For the neural network, the SHAP values and relative importance of TIME, SUNRISE, and SUNSET are zero, whereas SUNPERCENT retains a non-zero SHAP value. This suggests that the neural network effectively captures the information encoded in Equation 5, prioritizing SUNPERCENT as the primary contributor to CBLH predictions. In contrast, the ExtraTreesRegressor assigns reduced but non-zero relative importance to SUNRISE and SUNSET, indicating a broader distribution of feature contributions.

These differences likely stem from the distinct SHAP explainers used for each model. The ExtraTreesRegressor employs the TreeExplainer, which leverages the tree structure (split paths and leaf node values) to compute SHAP values directly, without requiring a background dataset. Conversely, the neural network uses the GradientExplainer, a local explanation method that relies on a background dataset (100 samples in this study) and computes SHAP values based on gradients near specific input points. When the local gradient for features such as TIME, SUNRISE, and SUNSET approaches zero, this reflects their negligible impact on the model's local decision boundary, resulting in corresponding SHAP values of zero. This explains the neural network's tendency to assign zero importance to these features, while the ExtraTreesRegressor's global approach captures their residual contributions.

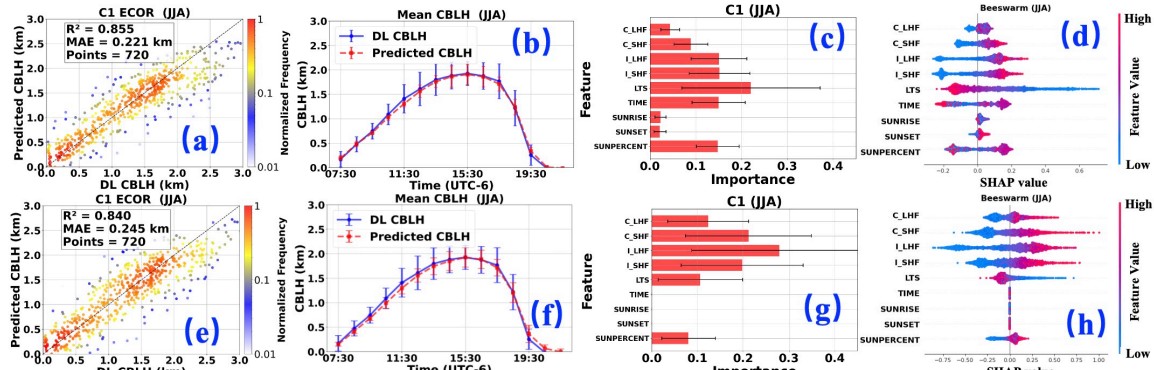

**Figure 9. Performance comparison of two machine learning frameworks during summer (JJA) (a-d) ExtraTreesRegressor: (a) R²
and MAE, (b) diurnal variations, (c) relative influence of input variables and (d) SHAP values of input variables. (e-h) Neural
network: (e) R² and MAE, (f) diurnal variations, (g) relative influence of input variables and (h) SHAP values of input variables.**

### 3.4.2 Comparative Analysis of SHAP Value Estimation Methods for AutoKeras Neural Networks

To validate the reliability of SHAP values and assess differences across computation methods, we compare the results of alternative SHAP explainers with those shown in Fig. 9g–h. The GradientExplainer, used for the AutoKeras neural network, approximates SHAP values by computing gradients of input features relative to model outputs, relying on a background dataset (100 samples in this study) to estimate feature contributions. The choice of background dataset can influence results, as GradientExplainer assumes local differentiability and quantifies feature importance based on gradient information. Consequently, features with near-zero gradients (e.g., those with minimal local influence near the background dataset) may yield zero SHAP values.





To mitigate this limitation, two additional explainers were employed: (1) KernelExplainer, a model-agnostic method that estimates SHAP values through sampling and weighted regression, suitable for any model. By sampling the global feature space, it captures non-local or nonlinear contributions, potentially yielding non-zero SHAP values even when local gradients are zero. However, KernelExplainer still requires a background dataset. (2) ExactExplainer, which does not require an explicit background dataset but uses a masking strategy, typically shap.maskers.Independent, to implicitly define the background distribution based on the data itself. By precisely computing Shapley values for all feature combinations, ExactExplainer provides the theoretically most accurate SHAP estimates, though it is computationally intensive.

Table 3. Performance Comparison of Various SHAP Value Calculation Methods.

| | SHAP Values | | |
| --- | --- | --- | --- |
| | GradientExplainer | KernelExplainer | ExactExplainer |
| C_LHF | 0.12 | 0.09 | 0.09 |
| C_SHF | 0.21 | 0.22 | 0.21 |
| I_LHF | 0.28 | 0.18 | 0.18 |
| I_SHF | 0.2 | 0.13 | 0.13 |
| LTS | 0.11 | 0.15 | 0.16 |
| TIME | 0 | 0.13 | 0.13 |
| SUNRISE | 0 | 0.02 | 0.02 |
| SUNSET | 0 | 0.01 | 0.01 |
| SUNPERCENT | 0.08 | 0.07 | 0.07 |

Table 3 summarizes the performance of different SHAP explainers. The GradientExplainer assigns zero SHAP values to input features with low influence, resulting in larger errors, but it offers high computational efficiency and requires fewer resources. In contrast, the ExactExplainer provides more reliable results but incurs high computational complexity, making it resource-intensive. For scenarios with limited computational resources and a need for high-accuracy SHAP values, the KernelExplainer is recommended as a balanced alternative. Notably, for features with lower relative importance, such as TIME, SUNRISE, and SUNSET, the ExactExplainer and KernelExplainer yield nearly identical results, with minor differences (approximately 0.01) observed for C_SHF and LTS.

Based on these findings, the SHAP value computation strategy in this study is as follows: For the TPOT-selected model (ExtraTreesRegressor), the TreeExplainer is used, leveraging its efficiency for tree-based models. For the AutoKeras neural network, the ExactExplainer is employed to compute SHAP values, as real-time computation is not required, prioritizing accuracy over speed.

## 3.5 Seasonal comparative analysis of Auto-ML's performance

### 3.5.1 Seasonal comparative analysis of Auto-ML's comprehensive performance

The performance of Auto-ML varies across different sites and seasons. As shown in Figure 10, after training with multi-year ECOR data at C1 sites, the model's performance is evaluated across four seasons. Figures 10a1–8a4 illustrate that autumn (SON) achieves the highest overall R² (0.860) but ranks second





in MAE (0.178 km). Winter (DJF) exhibits the lowest MAE (0.173 km) but the poorest R² (0.736). Summer (JJA) records the highest overall CBLH with strong performance (R²: 0.855, MAE: 0.221 km). Spring (MAM) yields an R² of 0.768 and an MAE of 0.239 km. Given the higher CBLH in summer and lower CBLH in winter, Auto-ML performs best overall in autumn.

### 3.5.2 Season-wise comparison of hourly averaged AutoML performance

However, when considering diurnal variations across seasons, summer (JJA) appears to perform best. As shown in Figures 10b1–9b4, predictions for spring (MAM) and autumn (SON) near the CBL top phase are approximately 0.1 km lower than DL observations. In winter (DJF), due to lower overall CBLH, predictions are about 0.05 km below observations. In contrast, summer (JJA) shows no significant
discrepancy near the CBL top (12:30–14:30), with predictions only slightly lower (~0.05 km) around 11:30. Potential reasons include: (1) a larger number of summer data points, leading Auto-ML to prioritize summer performance, and (2) distinct entrainment processes in summer compared to other seasons.

The entrainment process at the top of the atmospheric boundary layer exhibits a dual influence on
boundary layer development. When warm, dry air is entrained into the boundary layer, it enhances turbulent mixing and promotes vertical growth (Angevine et al., 1998). Conversely, if a strong inversion layer exists aloft, entrainment can suppress convection by dissipating turbulent kinetic energy and reducing upward heat flux (Lenschow et al., 2012). The entrainment rate $\omega_e$, defined as the volume flux of air drawn from the free atmosphere into the mixing layer per unit time at the boundary layer top, follows
a modified form of the classical entrainment parameterization (Lilly, 1968; Deardorff, 1979; Stull, 1988; Sullivan et al., 1998):

$$\omega_e = A \frac{\omega_*^3}{h \Delta \theta_v} \tag{6}$$

where, the entrainment efficiency coefficient A is around 0.2 (Beare et al., 2006; Cuxart et al., 2006); $\beta \approx 0.5$ is an empirical coefficient representing stratification suppression (Pino et al., 2003); h is CBLH,
$\omega_*$ is convective velocity scale(Deardorff, 1979); $\Delta \theta_v$ is virtual potential temperature jump (virtual potential temperature at the bottom of the inversion layer - average virtual potential temperature of the mixing layer); $\theta_{v0}$ is the surface virtual potential temperature. Equate 6 indicates that deeper boundary layers (larger h) exhibit weaker entrainment due to diminished turbulent energy (Beare et al., 2008). Additionally, when cloud fraction exceeds ~60%, the boundary layer growth rate declines by over 50%,
as cloud shading suppresses surface-driven turbulence (Zhang et al., 2020).

In humid summer conditions, high specific humidity (>18–20 g/kg) further inhibits boundary layer growth through multiple pathways: (a) increased cloudiness reduces surface solar heating (Luo et al., 2024), (b) precipitation depletes convective available potential energy (Hohenegger et al., 2013), and (c) evaporative
cooling enhances stability (Zhang et al., 2003). Observational studies confirm that tropical moist boundary layers are 30–40% shallower than their arid counterparts (von Engeln & Teixeira, 2013), highlighting moisture's threshold-like suppression effect.



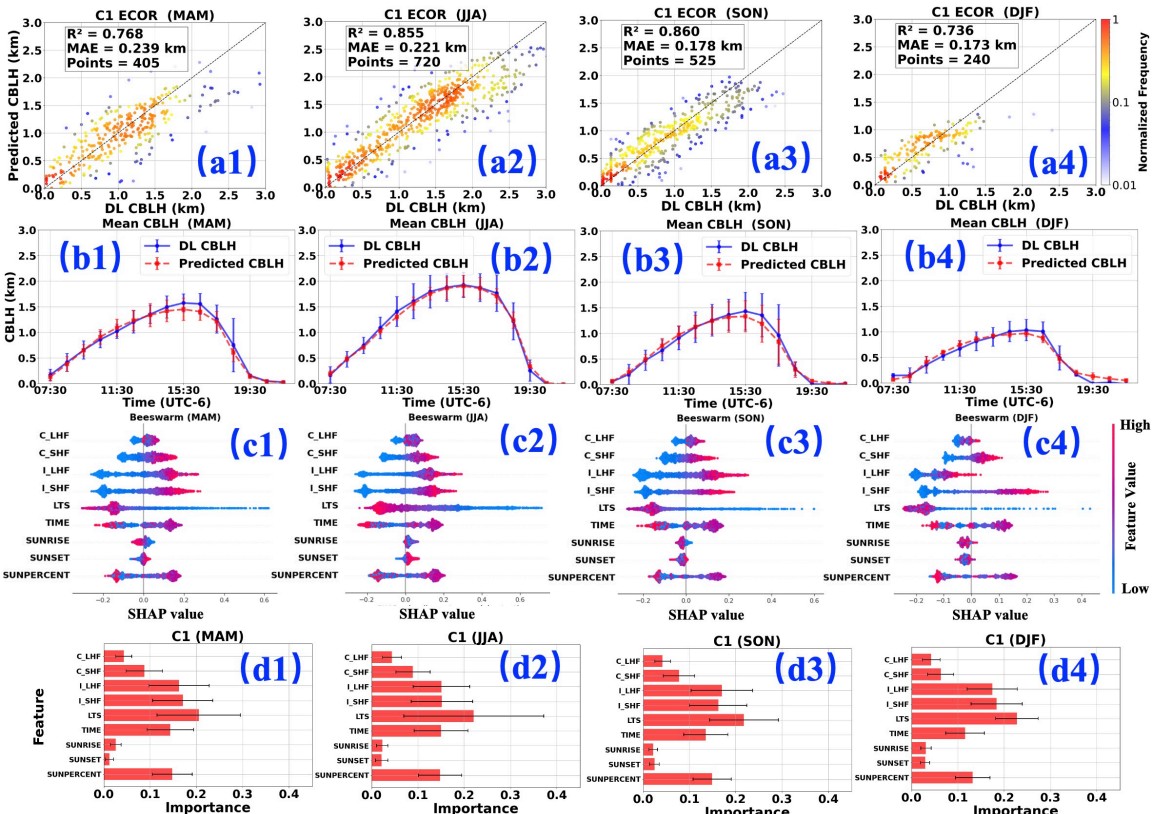

**Figure 10. Seasonal performance of the C1 ECOR model. The four columns represent the four seasons: spring (MAM), summer (JJA), autumn (SON), and winter (DJF); (a) R² and MAE metrics;(b) Diurnal evolution of mean CBLH; (c) Beeswarm (SHAP values); (d)SHAP-derived feature importance.**

Further analysis of Fig. 10b1–10b4 reveals that the interquartile range (IQR) error bars for predicted CBLH are consistently narrower than those for DL-derived CBLH. This is particularly evident in summer (JJA, Fig. 10b2), where, for instance, at 11:30, the IQR of DL CBLH is approximately four times that of the predicted values. This indicates that, despite the mean values of DL CBLH and predicted CBLH being similar (nearly overlapping at 11:30 with an error <0.01 km), the boundary layer development phase is governed by heat flux and LTS, which determine the attainable CBLH height. However, DL CBLH exhibits greater variability due to influences from additional meteorological factors, such as wind and low-level jets (LLJ). In contrast, the predicted CBLH, derived solely from thermodynamic parameters, lacks information on these factors, resulting in a more constrained range. A similar trend is observed in winter (DJF) with pronounced consistency. In spring and autumn, while a comparable pattern exists, the differences between predicted and observed values are smaller, suggesting lower variability (or complexity) in meteorological conditions compared to summer.

### 3.5.3 SHAP dependence between Auto-ML input parameters and CBLH

Analysis of Fig. 10c1–c4 reveals the relationships between various variables and CBLH. Initially, heat flux components (C_SHF, C_LHF, I_SHF, I_LHF) exhibit a predominantly positive correlation with



CBLH, indicating that stronger heat flux corresponds to greater CBLH. Conversely, LTS shows a negative correlation with CBLH, suggesting that higher LTS values are associated with reduced CBLH. Time and SUNPERCENT display a biphasic relationship with CBLH: a positive correlation is observed during the pre-peak phase, where CBLH increases with time, while a negative correlation emerges post-peak as CBLH decreases with time. This behavior is consistent with CBL dynamics. Additionally, SUNRISE exhibits a weak negative correlation with CBLH, implying that later sunrise times correspond to lower CBLH, whereas earlier sunrise times are linked to higher CBLH. Similarly, later sunset times are associated with higher CBLH, and earlier sunset times with lower CBLH. These patterns align well with the established development processes of the atmospheric boundary layer. Since SUNPERCENT integrates the effects of time, sunrise, and sunset, its relationship with CBLH closely mirrors that of time.

While the general relationships between input parameters and CBLH are outlined above, seasonal variations are notable. For instance, LTS consistently exhibits a negative correlation with CBLH in spring (MAM), autumn (SON), and winter (DJF). However, in summer, certain data points show a positive correlation, suggesting that under specific complex meteorological conditions in summer, factors beyond LTS dominate CBLH development, further highlighting the complexity of summer CBL dynamics. Additionally, I_LHF displays a positive correlation with CBLH across spring, summer, and autumn, but a negative correlation in winter, despite most corresponding SHAP values exceeding -0.1, creating a stark contrast with the other seasons. A similar, though less pronounced, trend is observed with C_SHF in winter. This phenomenon may be attributed to the dominance of northerly monsoon winds in winter, exacerbating cold, dry conditions. An increase in I_LHF could reduce I_SHF, potentially suppressing turbulence generation. In winter, both SUNRISE and SUNSET exhibit a negative correlation with CBLH, indicating that the overall CBLH remains low during this season. The specific mechanisms underlying these phenomena require further investigation.

### 3.5.4 Relative importance of input parameters in Auto-ML

The relative importance of different parameters across the four seasons exhibits a consistent pattern. LTS ranks highest (0.2–0.25), as it determines the energy required for CBLH growth. Next are the instantaneous heat flux components (I_SHF: 0.15–0.18 and I_LHF: 0.15–0.18), indicating that current heat flux plays a critical role in sustaining CBLH. Following these are TIME (0.12–0.15) and SUNPERCENT (0.12–0.15), which collectively govern the diurnal variation in boundary layer development; a positive correlation is observed before CBLH peaks (typically when SUNPERCENT is around 0.5), transitioning to a negative correlation post-peak. Subsequently, C_SHF (0.06–0.09) exceeds C_LHF (~0.05) in influence. The least impactful factors are SUNRISE (~0.02) and SUNSET (0.01–0.02). These relative importance values are consistent with the discussion in Section 3.5.3.

However, minor seasonal variations in the relative importance of these parameters are observed. For instance, in summer, LTS reaches a relative importance of 0.22, but its error bar extends to 0.3, suggesting that while LTS temporarily dominates CBLH development, its influence is significantly modulated by other conditions, underscoring the complexity of summer boundary layer dynamics. In contrast, during



winter, LTS maintains a relative importance of approximately 0.23 with an error bar of only 0.1, indicating its dominant and stable role in governing winter boundary layer development.

## 3.6 Comparative analysis of multi-site training with site-specific testing

Previous analyses employed Auto-ML models trained separately for individual sites. To investigate potential performance improvements, we conducted experiments using combined multi-site training followed by site-specific testing while controlling for cross-heat-flux interference. Two independent test groups were evaluated: the ECOR cluster (C1, E37, E39) and the EBBR cluster (C1, E32, E39), with comparative results presented in Table 4. Figure 10 further illustrates the CBLH diurnal variation patterns to enhance temporal resolution analysis.

Table 4. The results of multi-site training with site-specific testing

|  | ECOR | | | EBBR | | |
|---|---|---|---|---|---|---|
|  | C1 | E37 | E39 | C1 | E32 | E39 |
| R2 | 0.851 | 0.832 | 0.806 | 0.837 | 0.778 | 0.82 |
| MAE (km) | 0.198 | 0.19 | 0.211 | 0.203 | 0.249 | 0.205 |
| C_LHF | 0.04 | 0.03 | 0.03 | 0.07 | 0.06 | 0.07 |
| C_SHF | 0.11 | 0.11 | 0.11 | 0.03 | 0.04 | 0.04 |
| I_LHF | 0.09 | 0.09 | 0.09 | 0.15 | 0.14 | 0.15 |
| I_SHF | 0.17 | 0.18 | 0.18 | 0.14 | 0.13 | 0.15 |
| LTS | 0.2 | 0.19 | 0.18 | 0.2 | 0.24 | 0.18 |
| TIME | 0.16 | 0.16 | 0.16 | 0.18 | 0.17 | 0.17 |
| SUNRISE | 0.04 | 0.05 | 0.05 | 0.03 | 0.03 | 0.04 |
| SUNSET | 0.03 | 0.03 | 0.03 | 0.03 | 0.03 | 0.03 |
| SUNPERCENT | 0.16 | 0.16 | 0.17 | 0.17 | 0.16 | 0.17 |

To evaluate the performance of the ECOR and EBBR models, we trained both using 70% of the data from three sites combined and tested them on the remaining 30% of site-specific data. The results show that ECOR outperforms EBBR overall, with an average $R^2$ of 0.830 and MAE of 0.200 across the three sites, compared to EBBR's average $R^2$ of 0.812 and MAE of 0.219. However, at site E39, EBBR achieves a better $R^2$ (0.820) and MAE (0.205) than ECOR's $R^2$ (0.806) and MAE (0.211). The overall performance surpassed that of training and testing solely with C1 site data: ECOR ($R^2$: 0.851 vs. 0.845; MAE: 0.198 km vs. 0.207 km), EBBR ($R^2$: 0.837 vs. 0.834; MAE: 0.203 km vs. 0.205 km).

In terms of input parameter importance, ECOR exhibits minimal variation, with all parameters contributing approximately 0.01, except for LTS at 0.02. For EBBR, the LTS contribution at site E32 (0.24) is 0.06 higher than at E39 (0.18). Despite E32's lower overall performance ($R^2$: 0.778, MAE: 0.249) compared to E39 ($R^2$: 0.820, MAE: 0.205), E32's predictions are more accurate near the convective boundary layer top (~15:30), closely aligning with observations (mean values nearly overlap). In contrast, E39's predictions at the same time are approximately 0.1 km lower than observed. This suggests that E32's emphasis on LTS enhances prediction accuracy near the CBL top (Fig.10), where LTS is a critical factor.

The primary influencing factors for ECOR and EBBR were similar (in table 4), with the largest difference observed in LTS (E32: 0.24, E39: 0.18). Additionally, the allocation of SHF and LHF at E39 differed



significantly between ECOR and EBBR, with notable disparities in C_LHF (0.07 vs. 0.03) and C_SHF (0.03 vs. 0.04). However, other factors, such as LTS and SUNPERCENT, showed near-identical patterns. These findings highlight discrepancies in heat flux measurements between ECOR and EBBR. Such differences typically require data assimilation in traditional PBL schemes but can be mitigated through machine learning's nonlinear combinations, yielding comparable CBLH estimates. This approach could also facilitate future heat flux data assimilation.

The lower overall R² and MAE at E32, combined with its distinct LTS contribution, indicate that local meteorological conditions at E32 differ from those at the other sites. According to Tang (2019), E32 is surrounded primarily by pasture, unlike the seasonal crops and grasslands near C1 and E39, consistent with findings in Section 3.3. Including E32's data in training improves its performance (R²: 0.778, MAE: 0.249) compared to Section 3.3 (R²: 0.718, MAE: 0.271), highlighting the benefit of site-specific data in model training.

As shown in the diurnal variations across different sites (Fig. 11), noticeable discrepancies remain among them. For the ECOR measurements, the predicted CBL top heights are lower than the observations at the C1 and E39 sites, whereas at the E37 site, the predicted values exceed the observed ones. This suggests that the entrainment process near the CBL top at E37 differs from that at C1 and E39. In contrast, the EBBR-based results show that predictions at C1 and E32 closely match the observations near the CBL top throughout the day, while at E39, the model tends to underestimate the CBL top height. Overall, the discrepancy in EBBR-based estimates is smaller than that of ECOR, particularly near the CBL top. One possible explanation is that the E39, C1, and E32 sites lie along a southeast wind trajectory, potentially leading to more consistent boundary layer characteristics across these locations (Chu et al., 2025a).

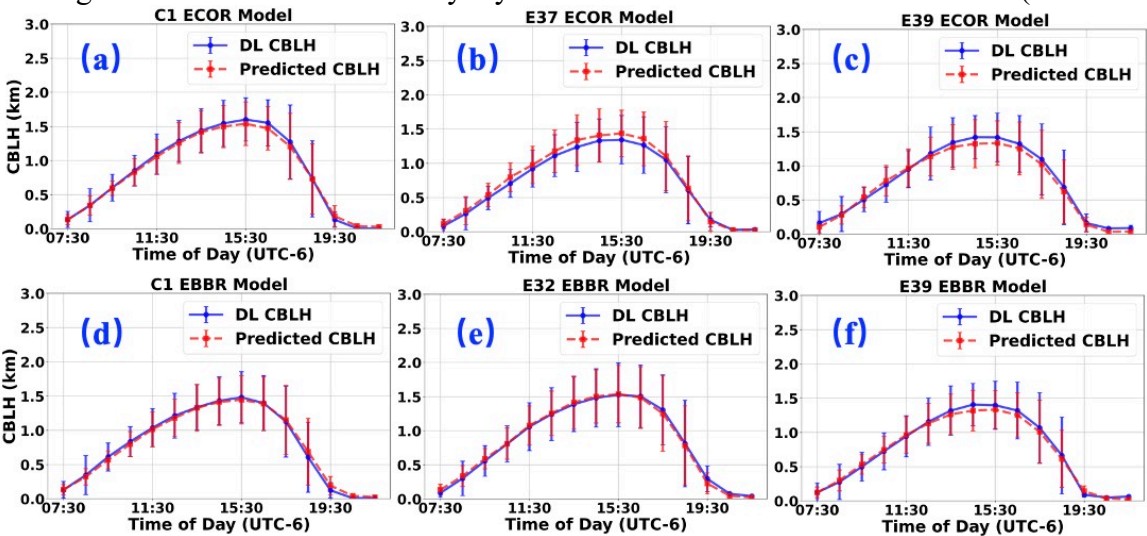

**Figure 11. Diurnal variation comparison of multi-site trained models evaluated through site-specific testing. Results are grouped by heat flux measurement system: (Top) ECOR sites (a) C1 (E14), (b) E37, (c) E39; (Bottom) EBBR sites (a) C1 (E13), (b) E32, (c) E39.**





## 4 Summary and conclusions

This study develops an Auto-ML framework for predicting CBLH, guided by thermodynamic physical constraints and the implicit diurnal cycle of CBLH. By leveraging TPOT and AutoKeras to automatically select optimal models, the approach bypasses manual comparisons of machine learning algorithms, enhancing efficiency and reproducibility. The resulting Auto-ML models, validated against Doppler lidar CBLH measurements, demonstrate robust performance with an overall $R^2$ of 0.84. Comparisons between ECOR and EBBR techniques for measuring surface heat and energy fluxes reveal consistent predictions, with an $R^2$ difference of approximately 0.011 and a MAE of 0.002 km. The models exhibit strong adaptability across multiple sites. When trained on ECOR data from the C1 site and applied to E37 and E39 sites within the ARM SGP network, the models achieve $R^2$ values of 0.787 and 0.806, respectively. Models trained on combined C1 and E39 data and tested on other sites show a gradual decline in $R^2$ and MAE with increasing distance yet maintain high predictive accuracy. These results underscore the transferability of the Auto-ML framework, highlighting its potential for integration with traditional numerical weather prediction models.

To enhance model interpretability, we conducted a comprehensive analysis of SHAP values using different explainers. The ExactExplainer was selected for its high accuracy in computing SHAP values for the AutoKeras neural network, despite its computational cost, as real-time computation was not required. In contrast, the TreeExplainer was employed for the TPOT-selected ExtraTreesRegressor, leveraging its efficiency for tree-based models. Seasonal performance at the C1 site was further evaluated, with a targeted comparison of model predictions during summer (JJA), revealing consistent feature importance patterns (e.g., dominance of LTS).

The study compared the performance of the C1 ECOR site across four seasons, revealing that autumn (SON) exhibited the best performance ($R^2$: 0.860, MAE: 0.178 km). This may be attributed to fewer clouds in autumn, reducing the influence of CBL top entrainment on CBLH development. Subsequently, the model was trained using data from multiple sites and tested individually. The overall performance surpassed that of training and testing solely with C1 site data: ECOR ($R^2$: 0.851 vs. 0.845; MAE: 0.198 km vs. 0.207 km), EBBR ($R^2$: 0.837 vs. 0.834; MAE: 0.203 km vs. 0.205 km). ECOR sites (C1, E32, E39) generally outperformed EBBR sites (C1, E37, E39) on average, though EBBR at E39 outperformed ECOR. The primary influencing factors for ECOR and EBBR were similar, with the largest difference observed in LTS (E32: 0.24, E39: 0.18). Additionally, the allocation of SHF and LHF at E39 differed significantly between ECOR and EBBR, with notable disparities in C_LHF (0.07 vs. 0.03) and C_SHF (0.03 vs. 0.04). However, other factors, such as LTS and SUNPERCENT, showed near-identical patterns. These findings highlight discrepancies in heat flux measurements between ECOR and EBBR. Such differences typically require data assimilation in traditional PBL schemes but can be mitigated through machine learning's nonlinear combinations, yielding comparable CBLH estimates. This approach could also facilitate future heat flux data assimilation. This implicit physically constrained Auto-ML approach significantly improves the accuracy and generalizability of CBLH predictions across diverse sites and seasons. By providing a scalable framework for boundary layer parameterization, it offers valuable




insights for refining atmospheric models and advancing the integration of machine learning in operational weather forecasting.

It should be noted that, although this study consistently refers to Auto-ML as "predicting" the CBLH, in the context of PBL schemes, it is more accurately described as "diagnosing" CBLH, given that the model uses a full day of data as both input and output. To enhance the model's applicability, it is critical to align it with the conventions of traditional PBL schemes by incorporating the CBLH output from the previous time step as an input for predicting the subsequent CBLH. Preliminary application of the model at the C1 site produces results consistent with those reported in this study ($R^2 = 0.822$; MAE = 0.199 km). The next step involves further optimization to meet additional requirements: extracting parameters from the CCPP-SCM PBL framework to predict the CBLH by Auto-ML, and then feeding this output back into the PBL framework to forecast the CBLH at the subsequent time step.

The Auto-ML PBL model has broad applications due to its accuracy and efficiency. It can support air quality forecasting by better predicting pollutant dispersion within the PBL, which is crucial for urban and industrial areas (Garratt, 1992; Stensrud, 2007). Additionally, its lightweight design makes it ideal for integration into local data-driven weather forecasting systems, providing accurate CBLH inputs to support low-altitude economic activities (Ben et al., 2024). The Auto-ML driven scalability further enables its use in data assimilation, integrating diverse observations for improved model initialization (Arcucci et al., 2021; Arcomano et al., 2023). As observational networks like ARM expand, this model offers a versatile tool for global atmospheric research.

The lightweight Auto-ML PBL model exhibits limitations in predicting peak MLH values near the PBL top, primarily due to two interrelated factors. First, its reliance on lidar-derived training data introduces uncertainties at higher altitudes, where a reduced SNR obscures sharp inversion layers. Second, Second, although the AutoML framework captures energy balance constraints to some extent, it does not fully represent critical physical processes such as entrainment dynamics and turbulence-driven mixing at the top of the CBL. These processes are especially important during the peak development phase of the boundary layer. At this stage, shear-driven turbulence and buoyancy fluxes play a dominant role in promoting vertical mixing and facilitating the entrainment of free-atmosphere air into the CBL. Without explicitly incorporating these mechanisms, the model may underrepresent key drivers of boundary layer growth, particularly under conditions of strong surface heating or elevated wind shear. This limitation highlights the need for physically informed hybrid models that can integrate data-driven approaches with boundary-layer process understanding. Unlike traditional schemes that parameterize these through TKE budgets or non-local mixing, the Auto-ML model lacks such dynamic constraints, reducing its sensitivity to abrupt inversion layer changes or synoptic-scale forcings (e.g., advective momentum fluxes) (Stevens, 2002; Cuxart et al., 2006; Fernando, 2010; Shin et al., 2021;). The IQR error bars for predicted CBLH are consistently narrower than those for DL-derived CBLH across all seasons, reflecting lower variability in predicted CBLH (based on thermodynamic parameters) compared to DL-derived CBLH, which is influenced by additional factors such as wind and low-level jets. Despite these limitations, a follow-up work is underway to develop an AI/ML-based emulator for parameterizing PBL, which will be applied into real-case simulations to assess its performance against conventional PBL schemes. Furthermore,



future enhancements may incorporate high-resolution observational data (e.g., uncrewed aircraft systems or airborne LiDAR) to directly sample the entrainment zone, thereby improving physical consistency. Additionally, integrating other boundary-layer parameters (e.g., turbulent dissipation rate; Chu et al., 2025b) could further refine the model, enabling predictions of additional PBL variables for a more comprehensive parameterization scheme.

## Code and Data Availability

ARM data are available at https://doi.org/10.2172/1253897 (ARM, 2016). The algorithm for convective boundary layer height determination is also accessible at https://doi.org/10.1364/OE.451728 (Chu et al., 2022a).

## Author Contributions

YC and HG conceptualized the study. YC, LX, GL, and HG curated the data. YC conducted the formal analysis. YC, LX, and HG developed the methodology. YC and HG implemented the software. ZW provided supervision. YC performed validation. YC, GL, and LX wrote the original draft. YC, GL, LX, MD, H.H.S., JZ, HG, and ZW contributed to writing, review, and editing.

## Competing Interests

The contact author has declared that none of the authors has any competing interests.

## Disclaimer

No disclaimer is applicable.

## Acknowledgements

This research was supported by the DOE Atmospheric System Research (ASR) program under grant DE-SC0020171 and the National Science Foundation (NSF) under grant AGS-1917693. Jun Zhang acknowledges support from NOAA under grant NA22OAR4050669D, the Office of Naval Research (ONR) MURI under grant N00014-24-1-2554, and the National Science Foundation (NSF) under awards 2228299 and 2211308.



**Financial Support**

No disclaimer is applicable.

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
