# Peer review of "Machine Learning Model for Inverting Convective Boundary Layer Height with Implicit Physical Constraints and Its Multi-Site Applicability"

_EGUsphere, 2025_

## Author Comment (AC1)

**RC1:**

The preprint describes a machine-learning model ("Auto-ML") trained to diagnose the convective boundary layer height (CBLH) evolution over one day. Generally, I think the choices described to add physical grounding to the ML model are well-motivated, though the paper description of them as providing 'implicit physical constraints' may be a bit of a reach. The paper would be much stronger if it included a baseline method of CBLH prediction; without one it lacks context for judging the Auto-ML skill.

Reply: We appreciate the reviewers' constructive feedback on the physical basis of the ML model. We agree that the term "implicit physical constraints" may be too broad and will revise it to "implicit thermodynamic physical constraints." To contextualize AutoML's capabilities, we have added the Linear Regression algorithm results as a benchmark in the supplement to highlight AutoML's performance and discussed in the revised manuscript.

**Specific comments:**

**1.** Simply including LTS and surface fluxes as inputs and using the full day of CBLH as targets does not guarantee that the ML model will learn the correct physical constraints. It is fair to say that these choices introduce more physical grounding into the ML problem setup, but I think that describing these as "Implicit physical constraints" in the title and section 2.5 is too far-reaching.

Reply: We agree that including LTS and surface fluxes as inputs with full-day CBLH targets does not ensure the ML model captures all physical constraints. Morning boundary layer growth is thermodynamically driven, while afternoon CBLH peaks involve entrainment, typically parameterized (~0.2) but lacking direct physical representation. We revised it as "implicit thermodynamic physical constraints" in the title and further discussed in Section 2.4.

**2.** My reading of the multisite analyses in 3.3 and 3.6 is that generalizing the model to different sites is limited by the flux input differences and site-specific differences and that training the ML model on the site it is to be used on is needed to achieve the best skill. This seems to contradict the abstract ("transferability across ARM Southern Great Plains sites... confirm the model's robustness").

Reply: We appreciate the reviewer's suggestion. The core finding of our study is that current ML models for CBLH prediction exhibit limited transferability across sites due to site-specific factors. By using on thermodynamics as a primary driver, our model achieves improved transferability across ARM Southern Great Plains sites, as stated in the abstract. However, model performance (R²) declines with distance, which is physically reasonable due to variations in non-thermodynamic factors such as terrain and vegetation, beyond just flux differences. To address your concern, we changed the abstract to "The model's generalizability across multiple sites at the ARM SGP site demonstrates its potential for transfer to greater distances, offering a scalable approach for enhancing boundary layer parameterization in atmospheric models", consistent with findings in Sections 3.3 and 3.6.

**3.** There is no baseline for comparison to assess how much skill the ML models are adding. I suggest including a simple baseline R2 and MAE as calculated using the training set mean CBLH target (over full time range, and also seasonal for that analysis) and including this baseline on the skill figures and tables. This would add context for how much of an improvement the AutoML model is providing.

Reply: Thank you for this highly constructive suggestion, which significantly enhances the article's readability. We have incorporated a Linear Regression algorithm as a baseline to demonstrate the extent of improvement provided by AutoML. For details, see Text S1 and S3 and Figures S1 and S3 in the Supporting Information.

**4.** In the description of input and output data in Sec 2.6, I would add a sentence explicitly stating the dimensionality of the input and output data. Related to this point, it sounds like aside from sunrise and sunset times, each input has (n\_timestamps\_in\_day) values in the full input vector. However, later in the interpretability section, single importance scores are given for each input, which confused me. Are SHAP values calculated for each timestamp of an input and averaged together? Please clarify in the text.

Reply: We appreciate the reviewer's feedback. We have added a sentence in Section 2.5 (section 2.6 in the first manuscript) explicitly stating the dimensionality of the input and output data. Indeed, aside from sunrise and sunset times, each input consists of n\_timestamps\_in\_day values in the full input vector. For the interpretability section, we clarify that SHAP values are calculated to reflect the relative importance of each input variable across the entire day, not at individual timestamps. We have revised the text to make this clear and avoid confusion in section 3.4.1.

**5.** What was the best model out of the set in table 2 chosen by the AutoML? This should be added to the text. Was it one of the two models in section 3.4? Did any other models in Table 2 also have comparably good skill, or were some significantly worse? Some discussion of the best performing architecture is warranted as could relate to the model's ability to generalize. e.g. one would expect a tree-based model to have difficulty generalizing as the output distribution is bounded by its training set.

Reply: Thank you for the reviewer's insightful suggestions. The best model selected by AutoML varied across different sites and even with different training data splits for the same site, making it inconsistent to highlight a single model. To avoid misleading readers, we treated the AutoML process as a black box and did not specify a single best model in the initial draft. We have now included the best model, which is the "ExtraTreesRegressor" from Section 3.4, as clarified in Section 3.1. In Table S1 (Original manuscript Table 2), many models exhibited comparable performance, with R² differences within 0.01. We have added a discussion in the revised manuscript noting that, with limited training data, tree-based models generally outperformed neural network architectures. AutoML underperforms in winter may relate to generalization challenges, as tree-based models' outputs are constrained by their training set.

**6.** The methods section should include some information about the computational resources used in training. This affects the space of model hyperparameters that can be explored by the Auto-ML algorithm. In particular the tree depth in the tree-based methods is directly related to the distribution of possible model outputs.

Reply: Thank you for the suggestion. We have added Section 2.3 to describe the computational resources used: "Windows 11 OS, Intel® Core™ i9-10900 CPU @ 2.8 GHz, 32 GB RAM." We agree that hyperparameters, such as tree depth in tree-based methods, influence the AutoML model search space and output distribution. However, a comprehensive exploration of hyperparameter tuning for each model would be computationally intensive and impractical. Therefore, this study employs default hyperparameters to facilitate model comparisons. Our related work (DOI:10.3390/rs17081399, 2025a) provides a detailed discussion on the effects of tree depth, learning rate, and number of estimators. These findings do not impact the generality of the results presented here.

7. In the interpretability section, there should be some discussion of whether the results were surprising or expected given prior knowledge of boundary layer processes. E.g. "In spring and autumn, while a comparable pattern exists, the differences between predicted and observed values are smaller, suggesting lower variability (or complexity) in meteorological conditions compared to summer." and "Potential reasons include:... distinct entrainment processes in summer compared to other seasons". I am not familiar with boundary layer processes, so for readers like me: Is it implied that it is already known that summer has lower variability in conditions and distinct entrainment processes, or are those the authors' hypotheses to explain their findings?

Reply: Thank you for your insightful and valuable feedback. We address this below and have incorporated a detailed discussion in Section 3.5.2 of the revised manuscript.

We fully concur that explicitly addressing whether observed patterns align with prior knowledge—or represent interpretive hypotheses—will aid readers unfamiliar with these processes, fostering a more accessible and rigorous discussion. The peak convective boundary layer height (CBLH) in summer (~2 km; Fig. 8b2) exceeds that in winter (~1 km), consistent with established literature. However, no prior studies have employed thermodynamic parameters to predict CBLH, rendering this approach novel. At the same time, we delineate our interpretations of summer-specific discrepancies—e.g., the pronounced widening of the interquartile range (IQR) in JJA, potentially driven by unmodeled wind-driven advection and enhanced entrainment from intense convective activity—as hypotheses grounded in process-based reasoning, rather than established consensus, to underscore the contributions of this work.

These revisions are complemented by the addition of a new panel (Figure 8c), which visually contrasts absolute and relative differences diurnally and seasonally, enabling clearer discernment of scale-dependent patterns and their implications—for instance, highlighting how relative discrepancies exceed 0.5 in autumn and winter mornings/evenings, while remaining below 0.1 during midday across seasons. We now emphasize that the winter-summer contrasts in CBLH scale and IQR are consistent with known seasonal forcings on boundary layer development, whereas the diurnal sensitivities and summer-specific variabilities represent novel insights, which we attribute to unresolved complex interactions like advection and entrainment.

To further guide future refinements, we propose incorporating parameters such as entrainment rates, tempuature and wind profiles to mitigate these gaps. We believe these enhancements not only directly address your query by distinguishing expected patterns from our proposed explanations but also elevate the manuscript's scientific depth.

We believe these clarifications and revisions strengthen the interpretability of our results and address the reviewer's concerns comprehensively. Thank you again for your constructive feedback, which has helped refine the manuscript.

**8.** I appreciate the breakdown of the results into the seasonal comparisons in section 3.5.2 and discussion of the physical processes affecting the CBLH and its variability. Here and in other sections, I think the writers did a good job of explaining how the physical processes involved in boundary layer changes might explain their findings.

Reply: Thank you for the positive feedback on Section 3.5.2 and our discussion of physical processes linking boundary layer dynamics to CBLH variability.

**9.** The readability would be greatly improved if the main text section related to importance/interpretability just focused on the main takeaway (LTS dominates) and left the rest to an appendix. Similarly for the section about ECOR vs EBBR flux results; I did not feel the findings were salient to the main points of the paper.

Reply: We appreciate the reviewer's feedback on readability. The main focus of the paper is not solely to highlight LTS dominance but to demonstrate the accuracy and multi-site applicability of thermodynamic implicit constraints for full-day CBLH predictions, including seasonal comparisons. The comparison of ECOR and EBBR fluxes addresses a key challenge in atmospheric science regarding data assimilation, a potential further goal of using ML in this study. To improve readability and emphasize the main theme, we have moved Previous article manuscript Sections 2.3, Sections 2.4, Sections 3.1.2 and 3.2.2, along with Table 2, Figures 5 and 7, to the supplementary materials.

Other comments:

**10.** Please define the variables in equation 4.

Reply: Added.

11. Hyperparameters for the ExtraTreesRegressor in Sec 3.4 should be provided.

Reply: Added.

**12.** Why is only JJA used in the comparison of the different ML methods in 3.4? Is it because the authors specifically wanted to study the season with higher DL-derived CBLH variability? Please clarify in the text.

Reply: We thank the reviewer for the comment. The choice of JJA in Section 3.4 was driven by its higher DL-derived CBLH variability and the greater availability of data, ensuring more robust results. We have added a clarification in the first paragraph of Section 3.4: "JJA was selected due to its higher DL-derived CBLH variability and larger data volume, enhancing result reliability."

**13**. Table 4: What is being shown in the rows labeled by the inputs? Feature importance? Please clarify in the caption.

Reply: Fixed. Yes, it is "Feature importance".

**14.** In the conclusion, L849 states the ML model "significantly improves the accuracy and generalizability of CBLH predictions across diverse sites and seasons." This ought to be edited as without a baseline for comparison, it is unclear this improvement is relative to.

Reply: We added the baseline. The statement on L849 has been revised to: "This implicit thermodynamic physically constrained Auto-ML approach selects the best-performing machine learning model based on the dataset, improving the accuracy and generalizability of CBLH predictions across diverse sites."

---

## Author Comment (AC2)

**RC2:**

Review of the article titled "Machine Learning model for inverting convective boundary layer height with implicit physical constraints and its multi-site applicability" by Chu and coauthors for publication in Atmos. Chem. Phys.

1. The authors have used boundary layer (BL) height from the doppler lidar (DL), surface fluxes from eddy correlation (ECOR) and energy balance Bowen Ratio (EBBR), and thermodynamic stability from Atmospheric Emitted Radiance Interferometer (AERI) to construct a machine learning (ML) model for predicting BL height. The data from ARM SGP site, and other ancillary sites around SGP have been used. The main premise of the paper is using the off-the-shelf interfaces like TPOT and AutoKeras for training and validation, thereby leaving AI to pick the ML model. After model identification, the authors have applied the model to predict BL height over different seasons and different sites. The article is relatively well-written and fits the journals scope. However, I believe that the article lacks physical depth, and could be improved. Much of the discussion is on simply adapting the data for TPOT and AutoKeras, which is not novel. The paper is also too long at this point, some of the discussion is more suitable for a dissertation rather than a paper. So, mentioning few things below that can improve the article further.

Reply: Thank you for your thorough review and constructive feedback. We revised the paper accordingly, which have significantly improved the manuscript's quality and alignment with ACP's submission standards. We acknowledge the concern regarding the lack of physical depth and the manuscript's length. To address this, we have streamlined the discussion by focusing on the novel aspects of the ML model application and moved less critical content, such as the introduction to machine learning and EBBR-related details, to the supplementary materials. Discussions on machine learning-related content are intended to facilitate better understanding for readers unfamiliar with machine learning. This revision enhances conciseness and clarity while maintaining the paper's focus on the use of the machine learning model for CBLH prediction across seasons and sites.

2. I like that you are trying to use some physical constraints as input parameters to improve the ML model. LTS, time, and sun parameters are a good start (Line 345). However, through previous research it has been shown that presence of elevated humidity above the BL can also affect the BL development through radiative effects, and same is true for high-level clouds. These effects in some part will be reflected in the ECOR fluxes, but with a time delay. In addition, wind speed, wind direction, wind shear and wind veer have also been shown to be very important. So maybe you

can include the following parameters in your input models, as they are also available at the ARM sites: wind speed, wind direction, wind shear, wind veer, surface upwelling and downwelling longwave and shortwave radiation. Surface meteorological variables will also be good to include. I understand that including cloud properties might be hard, but given the strong expertise of authors Deng, Xue and Wang, they can include ceilometer cloud fraction and base height in it. This might significantly improve the model, and the shapely analysis will tell which parameters are important. Thank you.

Reply: Thank you for your positive feedback and valuable suggestions. We agree that incorporating additional parameters such as wind speed, wind direction, wind shear, wind veer, surface longwave and shortwave radiation, and ceilometer-derived cloud fraction and base height could enhance the model's performance. This paper focused on measurements readily avaiable from both observations and model simulations. We have revised the manuscript's conclusion to reflect these future research goals and will use SHAP analysis to assess their importance. Thank you for your insightful recommendations.

3. The authors have used AutoKeras and TPOT for selecting the best model, which is great. Given the small amount of data used in this work, the tree-based models could also be employed in TPOT. It will be good if you can tell us what model and the associated hyperparameters was picked by these two frameworks. I cannot tell if they training was done online, and if so the batch sizes etc. The hyperparameters then should be scrutinized to understand whether any more improvement can be made. On the same topic, it will be good if the authors can mention if they explicitly or the two frameworks implicitly regularized (normalized, bias correct etc.) the input parameters. I assume the (Line 345) sunrise and sunset times and time variable could be normalized by 24.

Reply: Thank you for your insightful and valuable feedback. In the revised manuscript, we have detailed the models selected by AutoKeras and TPOT, along with their associated hyperparameters, in Section 3.4. All training was conducted locally, and we have included a detailed description of the hardware environment in Section 2.3. Regarding input parameter regularization, both frameworks implicitly normalize inputs, ensuring that explicit normalization of sunrise, sunset, and time variables by 24 does not affect the results. We have clarified this in the revised manuscript in Section 2.5 (section 2.6 in the first manuscript).

4. Figure 4 onwards: these are nice figures, but maybe you can add another panel showing the time evolution of the difference between DL CBLH and predicted CBLH. This will truly tell if the model accurately captures the daytime evolution.

Reply: Thank you for the excellent suggestion. Accordingly, we have incorporated figures illustrating the temporal evolution of the discrepancies between the DL-derived CBLH and the predicted CBLH into Figure 4c and Supplementary Figure S5.

5. Figure 10: The connection between convective boundary layer and surface fluxes is clear during summer months, but I cannot tell how it works in winter months. Can you please elaborate on the number of samples going into this figure, especially for the colder seasons. It is difficult to assess as to how accurate the surface fluxes might be when the environment is cold and the surface is frozen. Otherwise, you can control for it by using temperature and advection in your input parameters. Thank you.

Reply: Thank you for your insightful and valuable feedback. In Figure 10 (now Figure 8 in the revised manuscript), we have specified the winter (DJF) sample size as 240 points, one-third of the summer sample size (720 points). Winter performance is indeed the weakest, with an R2 of 0.736. We agree that assessing surface flux accuracy in cold, frozen conditions is challenging, but for clear and scattering cloud cases used in this study, the monthly mean sensible and latent heat fluxes are still up to ~200 and ~100 W/m2 in the winter (DJF). The consistent temporal evolutions of convective boundary layer and fluxes are still clear in winter at the SGP site. We have incorporated this discussion in Section 3.5.2 of the revised manuscript. In future work, we plan to include temperature and advection as input parameters to improve model performance.

Figure: Monthly mean temporal variations of sensible heat flux (SHF) and latent heat flux (LHF) at site C1. E13 (EBBR: Energy Balance Bowen Ratio system, for measuring sensible and latent heat fluxes via the Bowen ratio method) and E14 (ECOR: Eddy Correlation system, for measuring turbulent fluxes via the eddy covariance method) are both located within 1 km of the C1 site and can thus be regarded as co-located with C1.